# Unlocking Feature Visualization for Deeper Networks with Magnitude Constrained Optimization

**Thomas Fel**[⋆1,2,4], **Thibaut Boissin**[⋆2,3], **Victor Boutin**[⋆1,2], **Agustin Picard**[⋆2,3], **Paul Novello**[⋆2,3]
**Julien Colin**[1,5], **Drew Linsley**[1], **Tom Rousseau**[4], **Rémi Cadène**[1], **Lore Goetschalckx**[1],
**Laurent Gardes**[4], **Thomas Serre**[1,2]
[1]Carney Institute for Brain Science, Brown University
[2]Artificial and Natural Intelligence Toulouse Institute
[3]Institut de Recherche Technologique Saint-Exupery
[4]Innovation & Research Division, SNCF    [5]ELLIS Alicante, Spain.
{thomas_fel@brown.edu, thibaut.boissin@irt-saintexupery.com}

## Abstract

Feature visualization has gained substantial popularity, particularly after the influential work by Olah et al. in 2017, which established it as a crucial tool for explainability. However, its widespread adoption has been limited due to a reliance on tricks to generate interpretable images, and corresponding challenges in scaling it to deeper neural networks. Here, we describe **MACO**, a simple approach to address these shortcomings. The main idea is to generate images by optimizing the phase spectrum while keeping the magnitude constant to ensure that generated explanations lie in the space of natural images. Our approach yields significantly better results – both qualitatively and quantitatively – and unlocks efficient and interpretable feature visualizations for large state-of-the-art neural networks. We also show that our approach exhibits an attribution mechanism allowing us to augment feature visualizations with spatial importance. We validate our method on a novel benchmark for comparing feature visualization methods, and release its visualizations for all classes of the ImageNet dataset on 🍪 Lens.

Overall, our approach unlocks, for the first time, feature visualizations for large, state-of-the-art deep neural networks without resorting to any parametric prior image model.

## 1  Introduction

The field of Explainable Artificial Intelligence (XAI) [2, 3] has largely focused on characterizing computer vision models through the use of attribution methods [4, 5, 6, 7, 8, 9, 10, 11, 12, 13]. These methods aim to explain the decision strategy of a network by assigning an importance score to each input pixel (or group of input pixels [14, 15, 16]), according to their contribution to the overall decision. Such approaches only offer a partial understanding of the learned decision processes as they aim to identify the location of the most discriminative features in an image, the "where", leaving open the "what" question, *i.e.* the semantic meaning of those features. Recent work [17, 18, 19, 20, 21, 22] has highlighted the intrinsic limitations of attribution methods [23, 24, 25, 26], calling for the development of methods that provide a complementary explanation regarding the "what".

Feature visualizations provide a bridge to fill this gap via the generation of images that elicit a strong response from specifically targeted neurons (or groups of neurons). One of the simplest approaches uses gradient ascent to search for such an image. In the absence of regularization, this optimization is

---

⋆ The authors contributed equally.

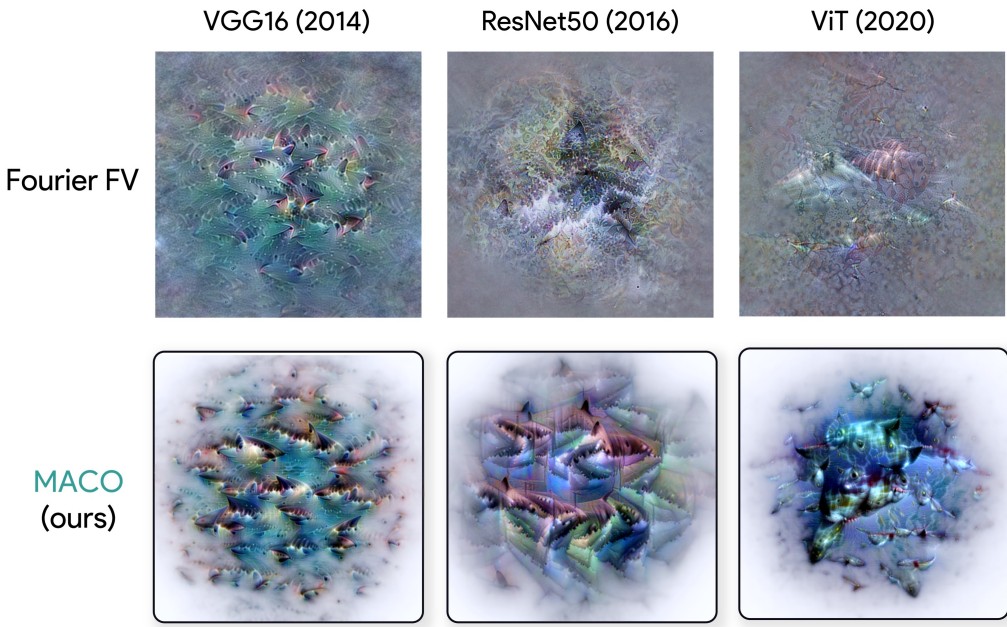

VGG16 (2014)    ResNet50 (2016)    ViT (2020)

Fourier FV

MACO (ours)

Figure 1: **Comparison between feature visualization methods for "White Shark" classification. (Top)** Standard Fourier preconditioning-based method for feature visualization [1]. **(Bottom)** Proposed approach, **MACO**, which incorporates a Fourier spectrum magnitude constraint.

known to yield highly noisy images – sometimes considered adversarial [27]. Hence, regularization methods are essential to rendering more acceptable candidate images. Such regularizations can consist of penalizing high frequencies in the Fourier domain [1, 28, 29, 30, 31], regularizing the optimization process with data augmentation [1, 32, 33, 34, 35, 36, 37] or restricting the search space to a subspace parameterized by a generative model [38, 39, 40, 41]. The first two approaches provide faithful visualizations, as they only depend on the model under study; unfortunately, in practice, they still fail on large modern classification models (*e.g.,* ResNet50V2 [42] and ViT [43], see Figure 1). The third approach yields interpretable feature visualizations even for large models but at the cost of major biases: in that case, it is impossible to disentangle the true contributions of the model under study from those of the generative prior model. Herein, we introduce a new feature visualization approach that is applicable to the largest state-of-the-art networks without relying on any parametric prior image model.

Our proposed approach, called MAgnitude Constrained Optimization (**MACO**), builds on the seminal work by Olah *et al.* who described the first method to optimize for maximally activating images in the Fourier space in order to penalize high-frequency content [1]. Our method is straightforward and essentially relies on exploiting the phase/magnitude decomposition of the Fourier spectrum, while exclusively optimizing the image's phase while keeping its magnitude constant. Such a constraint is motivated by psychophysics experiments that have shown that humans are more sensitive to differences in phase than in magnitude [44, 45, 46, 47, 48]. Our contributions are threefold:

  (*i*) We unlock feature visualizations for large modern CNNs without resorting to any strong parametric image prior (see Figure 1).
 (*ii*) We describe how to leverage the gradients obtained throughout our optimization process to combine feature visualization with attribution methods, thereby explaining both "what" activates a neuron and "where" it is located in an image.
(*iii*) We introduce new metrics to compare the feature visualizations produced with **MACO** to those generated with other methods.

As an application of our approach, we propose feature visualizations for FlexViT [49] and ViT [43] (logits and intermediate layers; see Figure 4). We also employ our approach on a feature inversion task to generate images that yield the same activations as target images to better understand what information is getting propagated through the network and which parts of the image are getting discarded by the model (on ViT, see Figure 6). Finally, we show how to combine our work with

a state-of-the-art concept-based explainability method [50] (see Figure 6b). Much like feature visualization, this method produces explanations on the semantic "what" that drives a model's prediction by decomposing the latent space of the neural network into a series of "directions" – denoted concepts. More importantly, it also provides a way to locate each concept in the input image under study, thus unifying both axes – "what" and "where". As feature visualization can be used to optimize in directions in the network's representation space, we employ **MACO** to generate concept visualizations, thus allowing us to improve the human interpretability of concepts and reducing the risk of confirmation bias. We showcase these concept visualizations on an interactive website: 🔮 Lens. The website allows browsing the most important concepts learned by a ResNet50 for all $1,000$ classes of ImageNet [42].

## 2   Related Work

Feature visualization methods involve solving an optimization problem to find an input image that maximizes the activation of a target element (neuron, layer, or whole model) [10]. Most of the approaches developed in the field fall along a spectrum based on how strongly they regularize the model. At one end of the spectrum, if no regularization is used, the optimization process can search the whole image space, but this tends to produce noisy images and nonsensical high-frequency patterns [51].

To circumvent this issue, researchers have proposed to penalize high-frequency in the resulting images – either by reducing the variance between neighboring pixels [28], by imposing constraints on the image's total variation [40, 41, 4], or by blurring the image at each optimization step [29]. However, in addition to rendering images of debatable validity, these approaches also suppress genuine, interesting high-frequency features, including edges. To mitigate this issue, a bilateral filter may be used instead of blurring, as it has been shown to preserve edges and improve the overall result [30]. Other studies have described a similar technique to decrease high frequencies by operating directly on the gradient, with the goal of preventing their accumulation in the resulting visualization [31]. One advantage of reducing high frequencies present in the gradient, as opposed to the visualization itself, is that it prevents high frequencies from being amplified by the optimizer while still allowing them to appear in the final image if consistently encouraged by the gradient. This process, known as "preconditioning" in optimization, can greatly simplify the optimization problem. The Fourier transform has been shown to be a successful preconditioner as it forces the optimization to be performed in a decorrelated and whitened image space [1]. The feature visualization technique we introduce in this work leverages a similar preconditioning. The emergence of high-frequency patterns in the absence of regularization is associated with a lack of robustness and sensitivity of the neural network to adversarial examples [27], and consequently, these patterns are less often observed in adversarially robust models [34, 33, 32]. An alternative strategy to promote robustness involves enforcing small perturbations, such as jittering, rotating, or scaling, in the visualization process [35], which, when combined with a frequency penalty [1], has been proved to greatly enhance the generated images.

Unfortunately, previous methods in the field of feature visualization have been limited in their ability to generate visualizations for newer architectures beyond VGG, resulting in a lack of interpretable visualizations for larger networks like ResNets [1]. Consequently, researchers have shifted their focus to approaches that leverage statistically learned priors to produce highly realistic visualizations. One such approach involves training a generator, like a GAN [40] or an autoencoder [52, 41], to map points from a latent space to realistic examples and optimizing within that space. Alternatively, a prior can be learned to provide the gradient (w.r.t the input) of the probability and optimize both the prior and the objective jointly [41, 30]. Another method involves approximating a generative model prior by penalizing the distance between output patches and the nearest patches retrieved from a database of image patches collected from the training data [38]. Although it is well-established that learning an image prior produces realistic visualizations, it is difficult to distinguish between the contributions of the generative models and that of the neural network under study. Hence, in this work, we focus on the development of visualization methods that rely on minimal priors to yield the least biased visualizations.

## 3   Magnitude-Constrained Feature Visualization

**Notations**   Throughout, we consider a general supervised learning setting, with an input space $\mathcal{X} \subseteq \mathbb{R}^{h \times w}$, an output space $\mathcal{Y} \subseteq \mathbb{R}^c$, and a classifier $\boldsymbol{f} : \mathcal{X} \to \mathcal{Y}$ that maps inputs $\boldsymbol{x} \in \mathcal{X}$ to a

prediction $y \in \mathcal{Y}$. Without loss of generality, we assume that $\boldsymbol{f}$ admits a series of $L$ intermediate spaces $\mathcal{A}_\ell \subseteq \mathbb{R}^{p_\ell}, 1 < \ell < L$. In this setup, $\boldsymbol{f}_\ell : \mathcal{X} \to \mathcal{A}_\ell$ maps an input to an intermediate activation $\boldsymbol{v} = (v_1, \ldots, v_{p_\ell})^\intercal \in \mathcal{A}_\ell$ of $\boldsymbol{f}$. We respectively denote $\mathcal{F}$ and $\mathcal{F}^{-1}$ as the 2-D Discrete Fourier Transform (DFT) on $\mathcal{X}$ and its inverse.

**Optimization Criterion.** The primary goal of a feature visualization method is to produce an image $\boldsymbol{x}^\star$ that maximizes a given criterion $\mathcal{L}_{\boldsymbol{v}}(\boldsymbol{x}) \in \mathbb{R}$; usually some value aggregated over a subset of weights in a neural network $\boldsymbol{f}$ (neurons, channels, layers, logits). A concrete example consists in finding a natural "prototypical" image $\boldsymbol{x}^\star$ of a class $k \in [\![1, K]\!]$ without using a dataset or generative models. However, optimizing in the pixel space $\mathbb{R}^{W \times H}$ is known to produce noisy, adversarial-like $\boldsymbol{x}^\star$ (see section 2). Therefore, the optimization is constrained using a regularizer $\mathcal{R} : \mathcal{X} \to \mathbb{R}^+$ to penalize unrealistic images:

$$\boldsymbol{x}^\star = \underset{\boldsymbol{x} \in \mathcal{X}}{\arg\max}\, \mathcal{L}_{\boldsymbol{v}}(\boldsymbol{x}) - \lambda \mathcal{R}(\boldsymbol{x}). \tag{1}$$

In Eq. 1, $\lambda$ is a hyperparameter used to balance the main optimization criterion $\mathcal{L}_{\boldsymbol{v}}$ and the regularizer $\mathcal{R}$. Finding a regularizer that perfectly matches the structure of natural images is hard, so proxies have to be used instead. Previous studies have explored various forms of regularization spanning from total variation, $\ell_1$, or $\ell_2$ loss [40, 41, 4]. More successful attempts rely on the reparametrization of the optimization problem in the Fourier domain rather than on regularization.

## 3.1 A Fourier perspective

Mordvintsev *et al.* [53] noted in their seminal work that one could use differentiable image parametrizations to facilitate the maximization of $\mathcal{L}_{\boldsymbol{v}}$. Olah *et al.* [1] proposed to reparametrize the images using their Fourier spectrum. Such a parametrization allows amplifying the low frequencies using a scalar $\boldsymbol{w}$. Formally, the prototypal image $\boldsymbol{x}^\star$ can be written as $\boldsymbol{x}^\star = \mathcal{F}^{-1}(\boldsymbol{z}^\star \odot \boldsymbol{w})$ with $\boldsymbol{z}^\star = \arg\max_{\boldsymbol{z} \in \mathbb{C}^{W \times H}} \mathcal{L}_{\boldsymbol{v}}(\mathcal{F}^{-1}(\boldsymbol{z} \odot \boldsymbol{w}))$.

Finding $\boldsymbol{x}^\star$ boils down to optimizing a Fourier buffer $\boldsymbol{z} = \boldsymbol{a} + i\boldsymbol{b}$ together with boosting the low-frequency components and then recovering the final image by inverting the optimized Fourier buffer using inverse Fourier transform.

However, multiple studies have shown that the resulting images are not sufficiently robust, in the sense that a small change in the image can cause the criterion $\mathcal{L}_{\boldsymbol{v}}$ to drop. Therefore, it is common to see robustness transformations applied to candidate images throughout the optimization process. In other words, the goal is to ensure that the generated image satisfies the criterion even if it is rotated by a few degrees or jittered by a few pixels. Formally, given a set of possible transformation functions – sometimes called augmentations – that we denote $\mathcal{T}$ such that for any transformation $\boldsymbol{\tau} \sim \mathcal{T}$, we have $\boldsymbol{\tau}(\boldsymbol{x}) \in \mathcal{X}$, the optimization becomes $\boldsymbol{z}^\star = \arg\max_{\boldsymbol{z} \in \mathbb{C}^{W \times H}} \mathbb{E}_{\boldsymbol{\tau} \sim \mathcal{T}}(\mathcal{L}_{\boldsymbol{v}}((\boldsymbol{\tau} \circ \mathcal{F}^{-1})(\boldsymbol{z} \odot \boldsymbol{w}))$.

Empirically, it is common knowledge that the deeper the models are, the more transformations are needed and the greater their magnitudes should be.

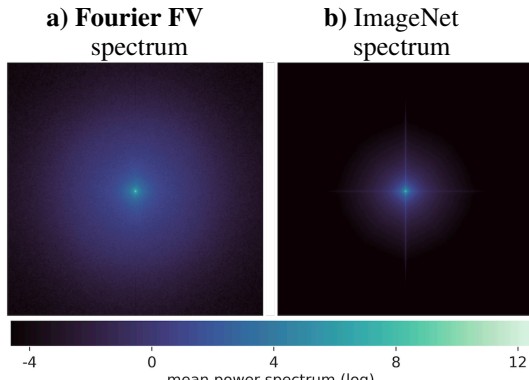

**a) Fourier FV spectrum**    **b) ImageNet spectrum**

-4    0    4    8    12
mean power spectrum (log)

Figure 2: **Comparison between Fourier FV and natural image power spectrum.** In **(a)**, the power spectrum is averaged over 10 different logits visualizations for each of the 1000 classes of ImageNet. The visualizations are obtained using the **Fourier FV**Fourier FV method to maximize the logits of a ViT network [1]. In **(b)** the spectrum is averaged over all training images of the ImageNet dataset.

To make their approach work on models like VGG, Olah *et al.* [1] used no less than a dozen transformations. However, this method fails for modern architectures, no matter how many transformations are applied. We argue that this may come from the low-frequency scalar (or booster) no longer working with models that are too deep. For such models, high frequencies eventually come through, polluting the resulting images with high-frequency content – making them impossible to interpret by humans. To empirically illustrate this phenomenon, we compute the $k$ logit visualizations obtained by maximizing each of the logits corresponding to the $k$ classes of a ViT using the parameterization used by Olah *et al.* In Figure 2 (left), we show the average of the

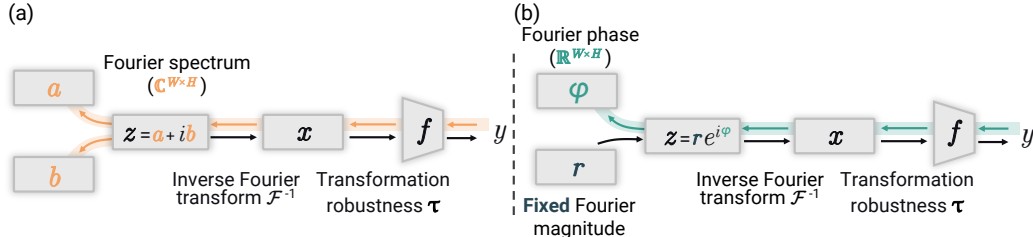

Figure 3: **Overview of the approach:** **(a)** Current Fourier parameterization approaches optimize the entire spectrum (yellow arrow). **(b)** In contrast, the optimization flow in our approach (green arrows) goes from the network activation ($y$) to the phase of the spectrum ($\varphi$) of the input image ($x$).

spectrum of these generated visualizations over all classes: $\frac{1}{k}\sum_{i=1}^{k}|\mathcal{F}(x_i^\star)|$. We compare it with the average spectrum of images on the ImageNet dataset (denoted $\mathcal{D}$): $\mathbb{E}_{x\sim\mathcal{D}}(|\mathcal{F}(x)|)$ (Figure 2, right panel). We observe that the images obtained through optimization put much more energy into high frequencies compared to natural images. Note that we did not observe this phenomenon in older models such as LeNet or VGG.

In the following section, we introduce our method named **MACO**, which is motivated by this observation. We constrain the magnitude of the visualization to a natural value, enabling natural visualization for any contemporary model, and reducing the number of required transformations to only two.

### 3.2 MACO: from Regularization to Constraint

Parameterizing the image in the Fourier space makes it possible to directly manipulate the image in the frequency domain. We propose to take a step further and decompose the Fourier spectrum $z$ into its polar form $z = re^{i\varphi}$ instead of its cartesian form $z = a + ib$, which allows us to disentangle the magnitude ($r$) and the phase ($\varphi$).

It is known that human recognition of objects in images is driven not by magnitude but by phase [44, 45, 46, 47, 48]. Motivated by this, we propose to optimize the phase of the Fourier spectrum while fixing its magnitude to a typical value of a natural image (with few high frequencies). In particular, the magnitude is kept constant at the average magnitude computed over a set of natural images (such as ImageNet), so $r = \mathbb{E}_{x\sim\mathcal{D}}(|\mathcal{F}(x)|)$. Note that this spectrum needs to be calculated only once and can be used at will for other tasks.

Therefore, our method does not backpropagate through the entire Fourier spectrum but only through the phase (Figure 3), thus reducing the number of parameters to optimize by half. Since the magnitude of our spectrum is constrained, we no longer need hyperparameters such as $\lambda$ or scaling factors, and the generated image at each step is naturally plausible in the frequency domain. We also enhance the quality of our visualizations via two data augmentations: random crop and additive uniform noise. To the best of our knowledge, our approach is the first to completely alleviate the need for explicit regularization –

---

**Algorithm 1 MACO**

**Require:** $r, \varphi_0 \sim \mathcal{U}([-\pi, \pi])^{W\times H}$
**Require:** $\alpha = 0$
    **for** $n \in [\![0, N]\!]$ **do**
        $\tau = \text{sample}(\mathcal{T})$
        $x_n = (\tau \circ \mathcal{F}^{-1})(re^{i\varphi_n})$
        $\varphi_{n+1} = \varphi_n + \eta\nabla_\varphi\mathcal{L}_v(x_n)$
        $\alpha = \alpha + |\nabla_{x_n}\mathcal{L}_v(x_n)|$
    **end for**
    **return** $x^* = \mathcal{F}^{-1}(re^{i\varphi_N}), \alpha$

---

using instead a hard constraint on the solution of the optimization problem for feature visualization. To summarize, we formally introduce our method:

**Definition 3.1 (MACO).** *The feature visualization results from optimizing the parameter vector $\varphi$ such that:*

$$\varphi^\star = \underset{\varphi\in\mathbb{R}^{W\times H}}{\arg\max}\, \mathbb{E}_{\tau\sim\mathcal{T}}(\mathcal{L}_v((\tau\circ\mathcal{F}^{-1})(re^{i\varphi})) \quad where \quad r = \mathbb{E}_{x\sim\mathcal{D}}(|\mathcal{F}(x)|)$$

*The feature visualization is then obtained by applying the inverse Fourier transform to the optimal complex-valued spectrum:* $x^\star = \mathcal{F}^{-1}((re^{i\varphi^\star})$

**Transparency for free:** Visualizations often suffer from repeated patterns or unimportant elements in the generated images. This can lead to readability problems or confirmation biases [54]. It is

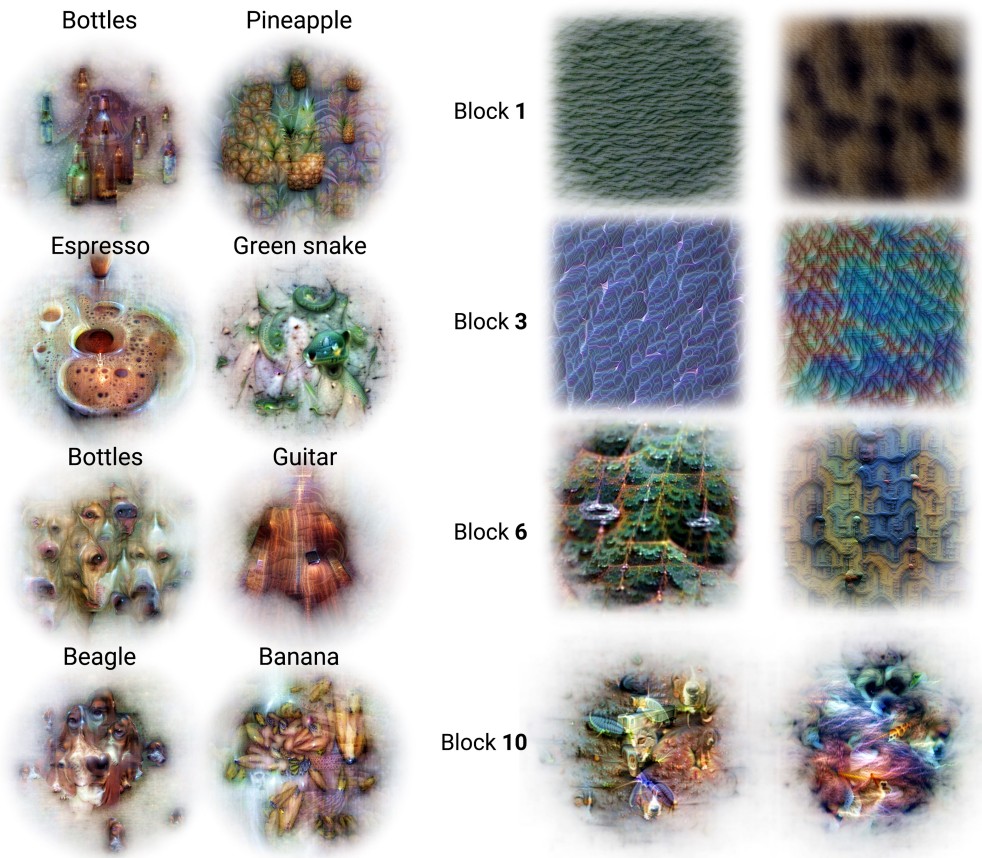

Figure 4: **(left) Logits and (right) internal representations of FlexiViT. MACO** was used to maximize the activations of **(left)** logit units and **(right)** specific channels located in different blocks of the FlexViT (blocks 1, 2, 6 and 10 from left to right).

important to ensure that the user is looking at what is truly important in the feature visualization. The concept of transparency, introduced in [53], addresses this issue but induces additional implementation efforts and computational costs.

We propose an effective approach, which leverages attribution methods, that yields a transparency map $\alpha$ for the associated feature visualization without any additional cost. Our solution shares theoretical similarities with SmoothGrad [5] and takes advantage of the fact that during backpropagation, we can obtain the intermediate gradients on the input $\partial \mathcal{L}_v(x)/\partial x$ for free as $\frac{\partial \mathcal{L}_v(x)}{\partial \varphi} = \frac{\partial \mathcal{L}_v(x)}{\partial x} \frac{\partial x}{\partial \varphi}$. We store these gradients throughout the optimization process and then average them, as done in SmoothGrad, to identify the areas that have been modified/attended to by the model the most during the optimization process. We note that a similar technique has recently been used to explain diffusion models [55]. In Algorithm 1, we provide pseudo-code for **MACO** and an example of the transparency maps in Figure 6 (third column).

## 4 Evaluation

We now describe and compute three different scores to compare the different feature visualization methods: Fourier (Olah *et al.*), CBR (optimization in the pixel space), and **MACO** (ours). It is important to note that these scores are only applicable to output logit visualizations. To keep a fair comparison, we restrict the benchmark to methods that do not rely on any learned image priors. Indeed, methods with learned prior will inevitably yield lower FID scores (and lower plausibility score) as the prior forces the generated visualizations to lie on the manifold of natural images.

**Plausibility score.** We consider a feature visualization plausible when it is similar to the distribution of images belonging to the class it represents. We quantify the plausibility through an OOD metric (Deep-KNN, recently used in [56]): it measures how far a feature visualization deviates from the corresponding ImageNet object category images based on their representation in the network's intermediate layers (see Table 1).

**FID score.** The FID quantifies the similarity between the distribution of the feature visualizations and that of natural images for the same object category. Importantly, the FID measures the distance between two distributions, while the plausibility score quantifies the distance from a sample to a distribution. To compute the FID, we used images from the ImageNet validation set and used the Inception v3 last layer (see Table 1). Additionally, we center-cropped our $512 \times 512$ images to $299 \times 299$ images to avoid the center-bias problem [39].

**Transferability score.** This score measures how consistent the feature visualizations are with other pre-trained classifiers. To compute the transferability score, we feed the obtained feature visualizations into 6 additional pre-trained classifiers (MobileNet [57], VGG16 [58], Xception [59], EfficientNet [60], Tiny ConvNext [61] and Densenet [62]), and we report their classification accuracy (see Table 2).

All scores are computed using 500 feature visualizations, each of them maximizing the logit of one of the ImageNet classes obtained on the FlexiViT [49], ViT[63], and ResNetV2[64] models. For the feature visualizations derived from Olah *et al.* [1], we used all 10 transformations set from the Lucid library*. CBR denotes an optimization in pixel space and using the same 10 transformations, as described in [29]. For **MACO**, $\tau$ only consists of two transformations; first we add uniform noise $\delta \sim \mathcal{U}([-0.1, 0.1])^{W \times H}$ and crops and resized the image with a crop size drawn from the normal distribution $\mathcal{N}(0.25, 0.1)$, which corresponds on average to 25% of the image. We used the NAdam optimizer [65] with $lr = 1.0$ and $N = 256$ optimization steps. Finally, we used the implementation of [1] and CBR which are available in the Xplique library [66] † which is based on Lucid.

| | FlexiViT | ViT | ResNetV2 |
|---|---|---|---|
| • **Plausibility score** (1-KNN) (↓) | | | |
| **MACO** | **1473** | **1097** | **1248** |
| Fourier [1] | 1815 | 1817 | 1837 |
| CBR [29] | 1866 | 1920 | 1933 |
| • **FID Score** (↓) | | | |
| **MACO** | **230.68** | **241.68** | **312.66** |
| Fourier [1] | 250.25 | 257.81 | 318.15 |
| CBR [29] | 247.12 | 268.59 | 346.41 |

| | FlexiViT | ViT | ResNetV2 |
|---|---|---|---|
| • **Transferability score**(↑): **MACO**/ Fourier [1] | | | |
| MobileNet | **68** / 38 | **48**/ 37 | **93** / 36 |
| VGG16 | **64** / 30 | **50** / 30 | **90** / 20 |
| Xception | **85** / 61 | **73** / 62 | **97** / 64 |
| Eff. Net | **88** / 25 | **63** / 25 | **82** / 21 |
| ConvNext | **96** / 52 | **84** / 55 | **93** / 60 |
| DenseNet | **84** / 32 | **66** / 31 | **93** / 25 |

Table 1: Plausibility and FID scores for different feature visualization methods applied on FlexiVIT, ViT and ResNetV2

Table 2: Transferability scores for different feature visualization methods applied on FlexiVIT, ViT and ResNetV2.

For all tested metrics, we observe that **MACO** produces better feature visualizations than those generated by Olah *et al.* [1] and CBR [29]. We would like to emphasize that our proposed evaluation scores represent the first attempt to provide a systematic evaluation of feature visualization methods, but we acknowledge that each individual metric on its own is insufficient and cannot provide a comprehensive assessment of a method's performance. However, when taken together, the three proposed scores provide a more complete and accurate evaluation of the feature visualization methods.

## 4.1 Human psychophysics study

Ultimately, the goal of any feature visualization method is to demystify the CNN's underlying decision process in the eyes of human users. To evaluate **MACO** 's ability to do this, we closely followed the psychophysical paradigm introduced in [67]. In this paradigm, the participants are presented with examples of a model's "favorite" inputs (i.e., feature visualization generated for a given unit) in addition to two query inputs. Both queries represent the same natural image, but have a

---

*https://github.com/tensorflow/lucid
†https://github.com/deel-ai/xplique

| FlexiViT | Plausibility ($\downarrow$) | FID ($\downarrow$) | logit magnitude ($\uparrow$) |
|---|---|---|---|
| MACO | 571.68 | 211.0 | 5.12 |
| - transparency | 617.9 (+46.2) | 208.1 (-2.9) | 5.05 (-0.1) |
| - crop | 680.1 (+62.2) | 299.2 (-91.1) | 8.18 (+3.1) |
| - noise | 707.3 (+27.1) | 324.5 (-25.3) | 11.7 (+3.5) |
| Fourier [1] | 673.3 | 259.0 | 3.22 |
| - augmentations | 735.9 (+62.6) | 312.5 (+53.5) | 12.4 (+9.2) |

Table 3: **Ablation study on the FlexiViT model:** This reveals that 1. augmentations help to have better FID and Plausibility scores, but lead to lesser salients visualizations (softmax value), 2. Fourier [1] benefits less from augmentations than MACO.

different part of the image hidden from the model by a square occludor. The task for participants is to judge which of the two queries would be "favored by the model" (i.e., maximally activate the unit). The rationale here is that a good feature visualization method would enable participants to more accurately predict the model's behavior. Here, we compared four visualization conditions (manipulated between subjects): Olah [1], MACO with the transparency mask (the transparency mask is decribed in 3.2), MACO without the transparency mask, and a control condition in which no visualizations were provided. In addition, the network (VGG16, ResNet50, ViT) was a within-subject variable. The units to be understood were taken from the output layer.

Based on the data of 174 participants on Prolific (www.prolific.com) [September 2023], we found both visualization and network to significantly predict the logodds of choosing the right query (Fig. 5). That is, the logodds were significantly higher for participants in both the MACO conditions compared to Olah. On the other hand, our tests did not yield a significant difference between Olah and the control condition, or between the two MACO conditions. Finally, we found that, overall, ViT was significantly harder to interpret than ResNet50 and VGG16, with no significant difference observed between the latter two networks. Full experiment and analysis details can be found in the supplementary materials, section C. Taken together, these results support our claim that even if feature visualization methods struggle in offering interpretable information as networks scale, MACO still convincingly helps people better understand deeper models while Olah's method [1] does not.

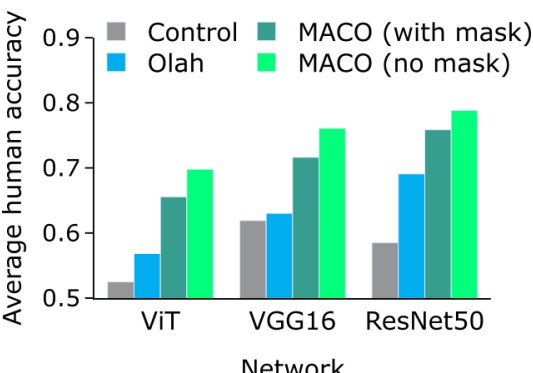

Figure 5: **Human causal understanding of model activations**. We follow the experimental procedure introduced in [67] to evaluate Olah and MACO visualizations on 3 different networks. The control condition is when the participant did not see any feature visualization.

## 4.2 Ablation study

To disentangle the effects of the various components of MACO, we perform an ablation study on the feature visualization applications. We consider the following components: (1) the use of a magnitude constraint, (2) the use of the random crop, (3) the use of the noise addition, and (4) the use of the transparency mask. We perform the ablation study on the FlexiViT model, and the results are presented in Table 3. We observe an inherent tradeoff between optimization quality (measured by logit magnitude) on one side, and the plausibility (and FID) scores on the other side. This reveals that plausible images which are close to the natural image distribution do not necessarily maximize the logit. Finally, we observe that the transparency mask does not significantly affect any of the scores confirming that it is mainly a post-processing step that does not affect the feature visualization itself.

# 5 Applications

We demonstrate the versatility of the proposed **MACO** technique by applying it to three different XAI applications:

**Logit and internal state visualization.** For logit visualization, the optimization objective is to maximize the activation of a specific unit in the logits vector of a pre-trained neural network (here a FlexiViT[49]). The resulting visualizations provide insights into the features that contribute the most to a class prediction (refer to Figure 4a). For internal state visualization, the optimization objective is to maximize the activation of specific channels located in various intermediate blocks of the network (refer to Figure 4b). This visualization allows us to better understand the kind of features these blocks – of a FlexiViT[49] in the figure – are sensitive to.

**Feature inversion.** The goal of this application is to find an image that produces an activation pattern similar to that of a reference image. By maximizing the similarity to reference activations, we are able to generate images representing the same semantic information at the target layer but without the parts of the original image that were discarded in the previous stages of the network, which allows us to better understand how the model operates. Figure 6a displays the images (second column) that match the activation pattern of the penultimate layer of a VIT when given the images from the first column. We also provide examples of transparency masks based on attribution (third column), which we apply to the feature visualizations to enhance them (fourth column).

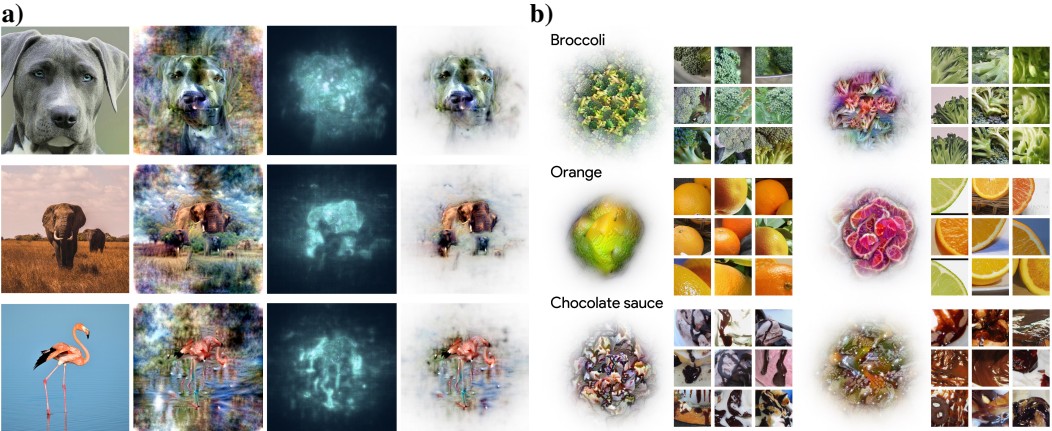

Figure 6: **Feature inversion and concept visualizaiton. a)** Images in the second column match the activation pattern of the penultimate layer of a ViT when fed with the images of the first column. In the third column, we show their corresponding attribution-based transparency masks, leading to better feature visualization when applied (fourth column). **b) MACO** is used to visualize concept vectors extracted with the CRAFT method [50]. The concepts are extracted from a ResNet50 trained on ImageNet. All visualizations for all ImageNet classes are available at ⊛ Lens.

**Concept visualization.** Herein we combine **MACO** with concept-based explainability. Such methods aim to increase the interpretability of activation patterns by decomposing them into a set of concepts [68]. In this work, we leverage the CRAFT concept-based explainability method [50], which uses Non-negative Matrix Factorization to decompose activation patterns into main directions – that are called concepts –, and then, we apply **MACO** to visualize these concepts in the pixel space. To do so, we optimize the visualization such that it matches the concept activation patterns. In Figure 6b, we present the top 2 most important concepts (one concept per column) for five different object categories (one category per row) in a ResNet50 trained on ImageNet. The concepts' visualizations are followed by a mosaic of patches extracted from natural images: the patches that maximally activate the corresponding concept.

# 6 Limitations

We have demonstrated the generation of realistic explanations for large neural networks by imposing constraints on the magnitude of the spectrum. However, it is important to note that generating realistic images does not necessarily imply effective explanation of the neural networks. The metrics introduced in this paper allow us to claim that our generated images are closer to natural images in latent space, that our feature visualizations are more plausible and better reflect the original distribution. However, they do not necessarily indicate that these visualizations helps humans in effectively communicating with the models or conveying information easily to humans. Furthermore, in order for a feature visualization to provide informative insights about the model, including spurious features, it may need to generate visualizations that deviate from the spectrum of natural images. Consequently, these visualizations might yield lower scores using our proposed metrics. Simultaneously, several interesting studies have highlighted the weaknesses and limitations of feature visualizations [54, 69, 67]. One prominent criticism is their lack of interpretability for humans, with research demonstrating that dataset examples are more useful than feature visualizations in understanding convolutional neural networks (CNNs) [54]. This can be attributed to the lack of realism in feature visualizations and their isolated use as an explainability technique. With our approach, MACO , we take an initial step towards addressing this limitation by introducing magnitude constraints, which lead to qualitative and quantitative improvements. Additionally, through our website, we promote the use of feature visualizations as a supportive and complementary tool alongside other methods such as concept-based explainability, exemplified by CRAFT [50]. We emphasize the importance of feature visualizations in combating confirmation bias and encourage their integration within a comprehensive explainability framework.

# 7 Conclusions

In this paper, we introduced a novel approach, MACO, for efficiently generating feature visualizations in modern deep neural networks based on (i) a hard constraint on the magnitude of the spectrum to ensure that the generated visualizations lie in the space of natural images, and (ii) a new attribution-based transparency mask to augment these feature visualizations with the notion of spatial importance. This enhancement allowed us to scale up and unlock feature visualizations on large modern CNNs and vision transformers without the need for strong – and possibly misleading – parametric priors. We also complement our method with a set of three metrics to assess the quality of the visualizations. Combining their insights offers a way to compare the techniques developed in this branch of XAI more objectively. We illustrated the scalability of MACO  with feature visualizations of large models like ViT, but also feature inversion and concept visualization. Lastly, by improving the realism of the generated images without using an auxiliary generative model, we supply the field of XAI with a reliable tool for explaining the semantic ("what" information) of modern vision models.

## Acknowledgments

This work was conducted as part the DEEL[‡] project. It was funded by ONR grant (N00014-19-1-2029), NSF grant (IIS-1912280 and EAR-1925481), and the Artificial and Natural Intelligence Toulouse Institute (ANITI) grant #ANR19-PI3A-0004. The computing hardware was supported in part by NIH Office of the Director grant #S10OD025181 via the Center for Computation and Visualization (CCV) at Brown University. J.C. has been partially supported by Intel Corporation, a grant by Banco Sabadell Foundation and funding from the Valencian Government (Conselleria d'Innovació, Industria, Comercio y Turismo, Direccion General para el Avance de la Sociedad Digital) by virtue of a 2023 grant agreement (convenio singular 2023).

---

[‡]https://www.deel.ai/

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
