[58] Karen Simonyan and Andrew Zisserman. Very deep convolutional networks for large-scale image recognition. *arXiv preprint arXiv:1409.1556*, 2014.

[59] François Chollet. Xception: Deep learning with depthwise separable convolutions. In *Proceedings of the IEEE conference on computer vision and pattern recognition*, pages 1251–1258, 2017.

[60] Mingxing Tan and Quoc Le. Efficientnet: Rethinking model scaling for convolutional neural networks. In *International conference on machine learning*, pages 6105–6114. PMLR, 2019.

[61] Zhuang Liu, Hanzi Mao, Chao-Yuan Wu, Christoph Feichtenhofer, Trevor Darrell, and Saining Xie. A convnet for the 2020s. In *Proceedings of the IEEE/CVF Conference on Computer Vision and Pattern Recognition*, pages 11976–11986, 2022.

[62] Gao Huang, Zhuang Liu, Laurens Van Der Maaten, and Kilian Q Weinberger. Densely connected convolutional networks. In *Proceedings of the IEEE conference on computer vision and pattern recognition*, pages 4700–4708, 2017.

[63] Alexander Kolesnikov, Lucas Beyer, Xiaohua Zhai, Joan Puigcerver, Jessica Yung, Sylvain Gelly, and Neil Houlsby. Big transfer (bit): General visual representation learning. In *Computer Vision – ECCV 2020: 16th European Conference, Glasgow, UK, August 23–28, 2020, Proceedings, Part V*, page 491–507, Berlin, Heidelberg, 2020. Springer-Verlag.

[64] Kaiming He, Xiangyu Zhang, Shaoqing Ren, and Jian Sun. Identity mappings in deep residual networks. In *Computer Vision–ECCV 2016: 14th European Conference, Amsterdam, The Netherlands, October 11–14, 2016, Proceedings, Part IV 14*, pages 630–645. Springer, 2016.

[65] Timothy Dozat. Incorporating nesterov momentum into adam. *Proceedings of the International Conference on Learning Representations (ICLR)*, 2016.

[66] Thomas Fel, Lucas Hervier, David Vigouroux, Antonin Poche, Justin Plakoo, Remi Cadene, Mathieu Chalvidal, Julien Colin, Thibaut Boissin, Louis Bethune, Agustin Picard, Claire Nicodeme, Laurent Gardes, Gregory Flandin, and Thomas Serre. Xplique: A deep learning explainability toolbox. *Workshop on Explainable Artificial Intelligence for Computer Vision (CVPR)*, 2022.

[67] Roland S Zimmermann, Judy Borowski, Robert Geirhos, Matthias Bethge, Thomas Wallis, and Wieland Brendel. How well do feature visualizations support causal understanding of cnn activations? *Advances in Neural Information Processing Systems*, 34:11730–11744, 2021.

[68] Amirata Ghorbani, James Wexler, James Y Zou, and Been Kim. Towards automatic concept-based explanations. *Advances in Neural Information Processing Systems*, 32, 2019.

[69] Robert Geirhos, Roland S. Zimmermann, Blair Bilodeau, Wieland Brendel, and Been Kim. Don't trust your eyes: on the (un)reliability of feature visualizations, 2023.

[70] J. Deng, W. Dong, R. Socher, L.-J. Li, K. Li, and L. Fei-Fei. ImageNet: A Large-Scale Hierarchical Image Database. In *Proceedings of the IEEE Conference on Computer Vision and Pattern Recognition (CVPR)*, 2009.

# A Additional results

In this section, we provide additional results for logit and internal feature visualizations, and feature inversion.

For all of the following visualizations, we used the same parameters as in the main paper. For the feature visualizations derived from Olah *et al.* [1], we used all 10 transformations set from the Lucid library[§]. For **MACO**, $\tau$ only consists of two transformations; first we add uniform noise $\delta \sim \mathcal{U}([-0.1, 0.1])^{W \times H}$ and crops and resized the image with a crop size drawn from the normal distribution $\mathcal{N}(0.25, 0.1)$, which corresponds on average to 25% of the image. We used the NAdam optimizer [65] with a $lr = 1.0$ and $N = 256$ optimization steps. Finally, we used the implementation of [1] and CBR which are available in the Xplique library [66][¶] which is based on Lucid.

## A.1 Logit and Internal State Visualization

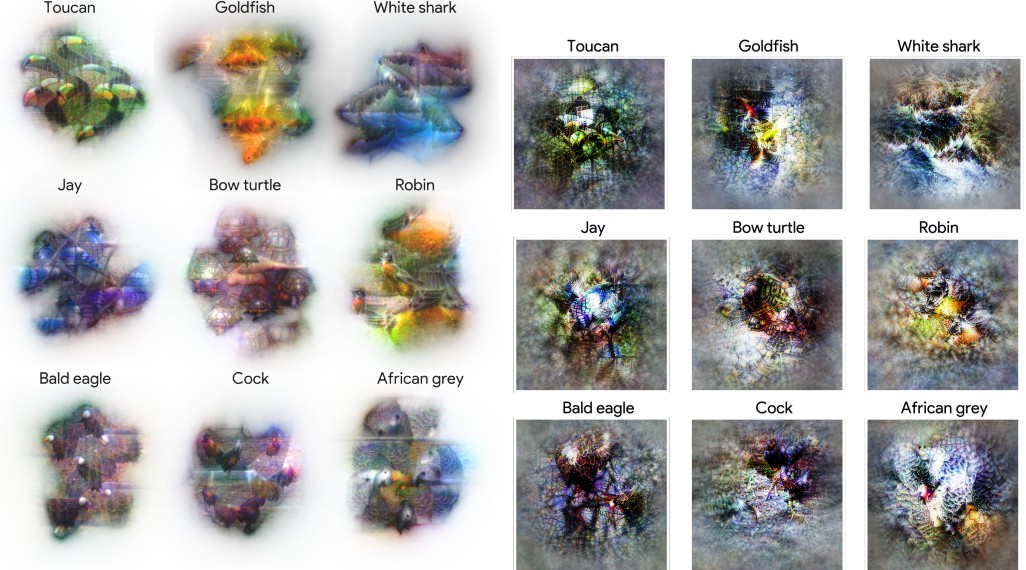

Figure S1: **Feature visualizations on a ResNet50.** We compare feature visualizations for 9 different classes from ImageNet generated via **MACO** and Olah *et al.*. We observe that our visualizations are much sharper, and parts of the target class show up really clearly. On the other hand, the baseline produces overexposed images with mostly textures that are somewhat related to the class.

---

[§]https://github.com/tensorflow/lucid
[¶]https://github.com/deel-ai/xplique

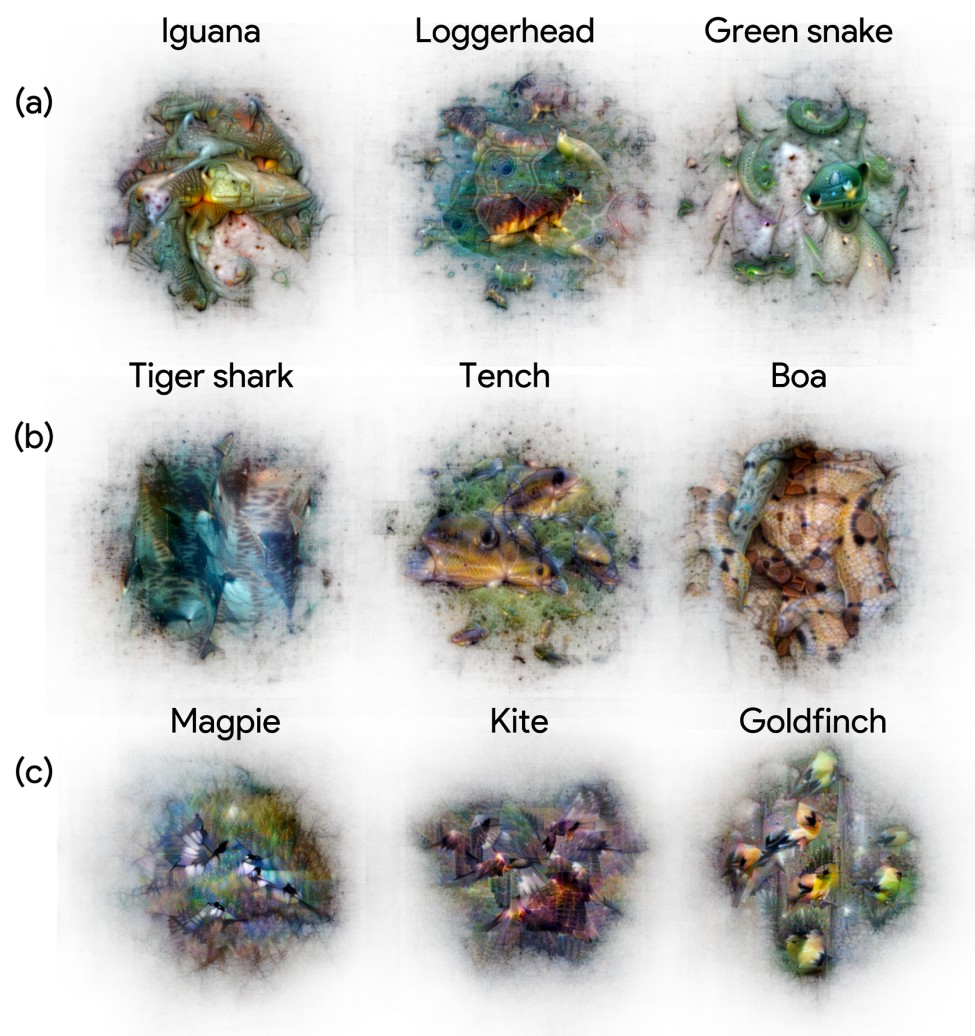

Figure S2: **Feature visualizations on FlexiViT, ViT and ResNet50.** We compare the feature visualizations from **MACO** generated for **(a)** FlexiViT, **(b)** ViT and **(c)** ResNet50 on a set of different classes from ImageNet. We observe that the visualizations get more abstract as the complexity of the model increases.

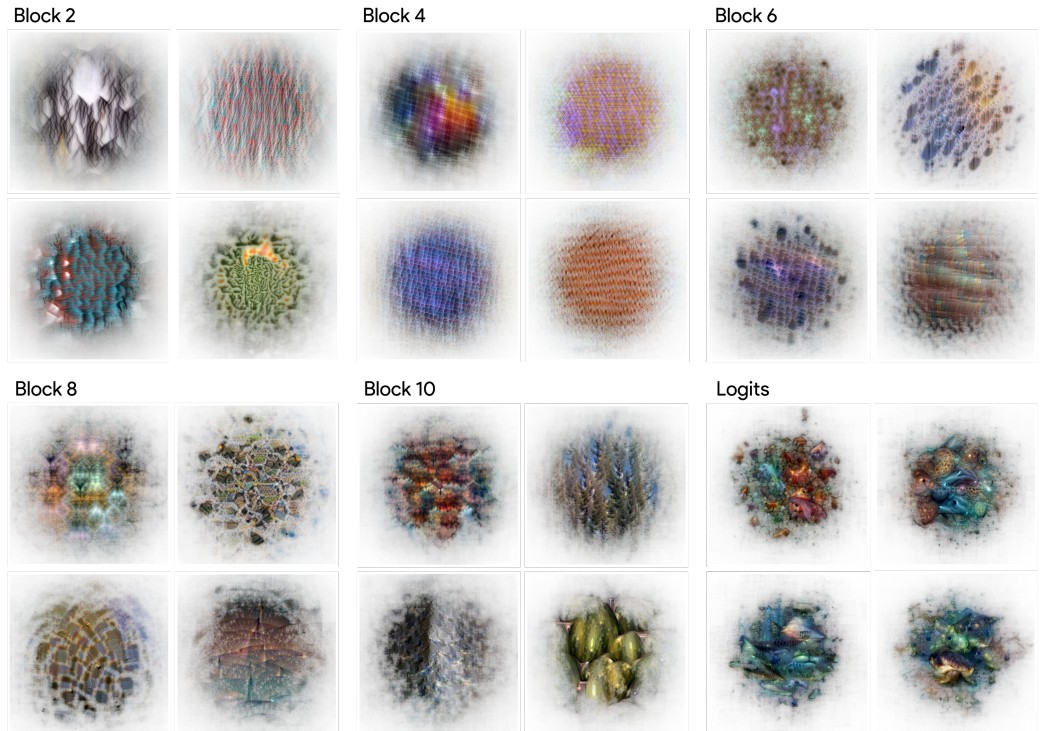

Figure S3: **Logits and internal representation of a ViT.** Using **MACO**, we maximize the activations of specific channels in different blocks of a ViT, as well as the logits for 4 different classes.

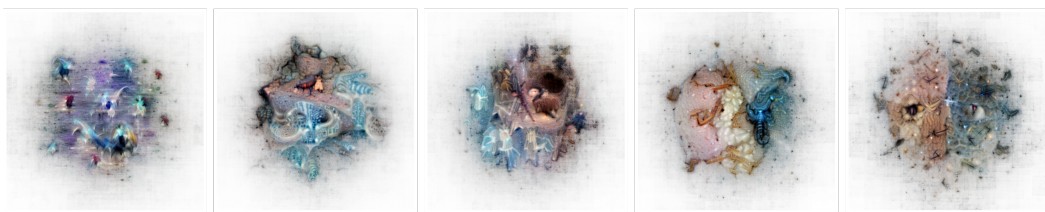

Figure S4: **Hue invariance.** Through feature visualization, we are able to determine the presence of hue invariance on our pre-trained ViT model manisfesting itself through phantom objects in them. This can be explained the data-augmentation that is typically employed for training these models.

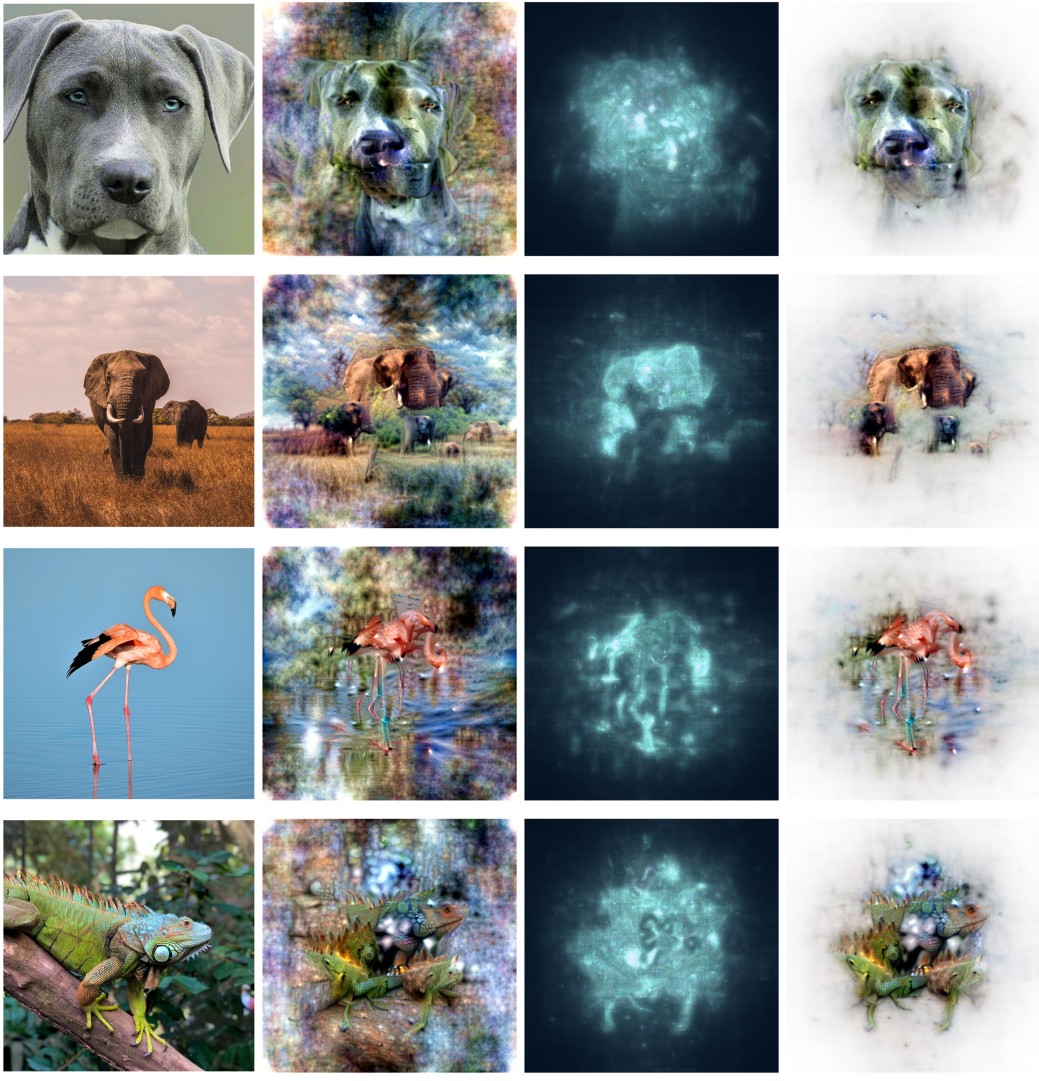

Figure S5: **Feature inversion and Attribution-based transparency.** We performed feature inversion on the images on the first column to obtain the visualizations (without transparency) on the second column. During the optimization procedure, we saved the intensity of the changes to the image in pixel space, which we showcase on the third column, we used this information to assign a transparency value, as exhibited in the final column.

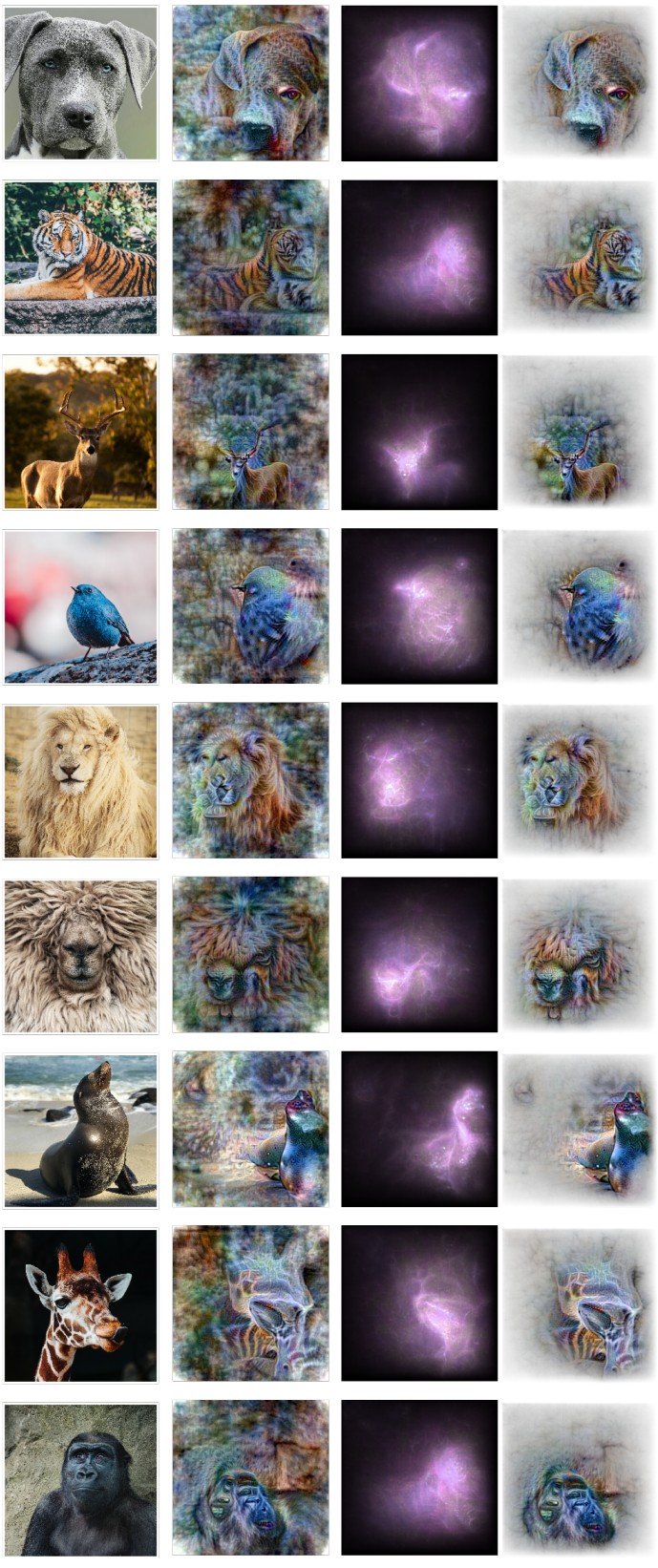

Figure S6: **Feature inversion and Attribution-based transparency.** We performed feature inversion on the images on the first column to obtain the visualizations (without transparency) on the second column. During the optimization procedure, we saved the intensity of the changes to the image in pixel space, which we showcase on the third column, we used this information to assign a transparency value, as exhibited in the final column.

## B    Screenshots from the website

For our website, we picked a ResNet50V2 that had been pre-trained on ImageNet [70] and applied CRAFT [50] to reveal the concepts that are driving its predictions for each class. CRAFT is a state-of-the-art, concept-based explainability technique that, through NMF matrix decompositions, allows us to factorize the networks activations into interpretable concepts. Furthermore, using Sobol indices and implicit differentiation, we can measure the importance of each concept, and trace its presence back to and locate it in the input image.

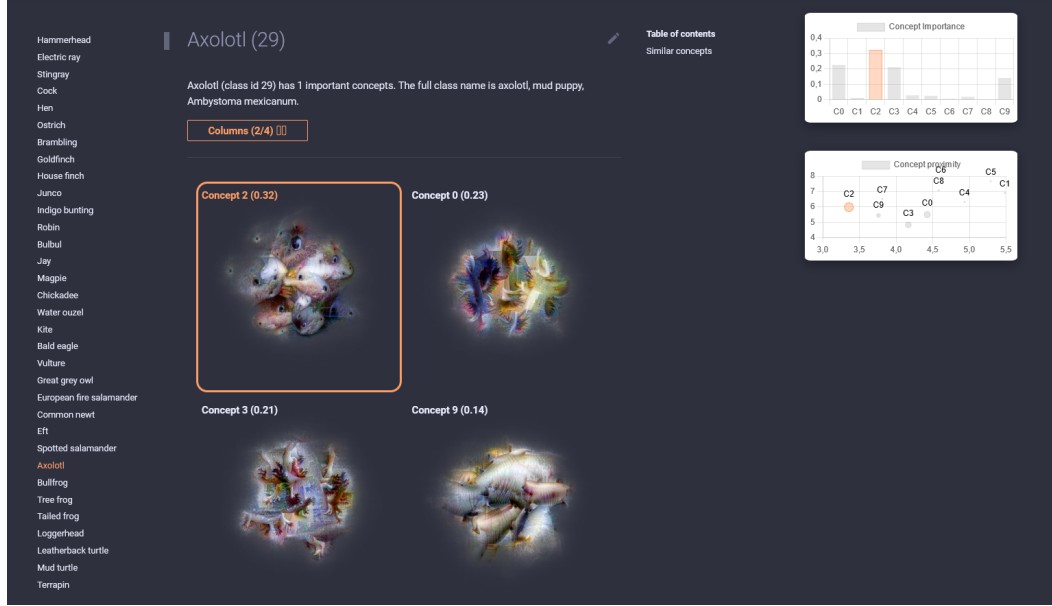

Figure S7: **Overview of a class explanation.** Screenshot of the first glimpse of the explanations for the class *axolotl*. We display the concept visualizations by order of importance (the most important concepts first), with a bar plot of the importance scores and a plot of their similarity.

In particular, we applied this technique to explain all the classes in ImageNet, and used MACO to generate visualizations that maximize the angle of superposition with each direction in the network's activation space (i.e. each concept). We also plot each of the concepts' global importance, as well as the similarities between concepts of the same class. For each class, we first showcase the most important concepts with their respective visualizations (see Fig. S7 and Fig. S8), and by clicking on them, we exhibit the crops that align the most with the concept (see Fig. S9). This allows us to diminish the effect of potential confirmation bias by providing two different approaches to understanding what the model has encoded in that concept.

Finally, we have also computed the similarity between concepts of different classes and display the closest (see Fig. S10). This feature can help better understand erroneous predictions, as the model may sometimes leverage similar concepts of different classes to classify.

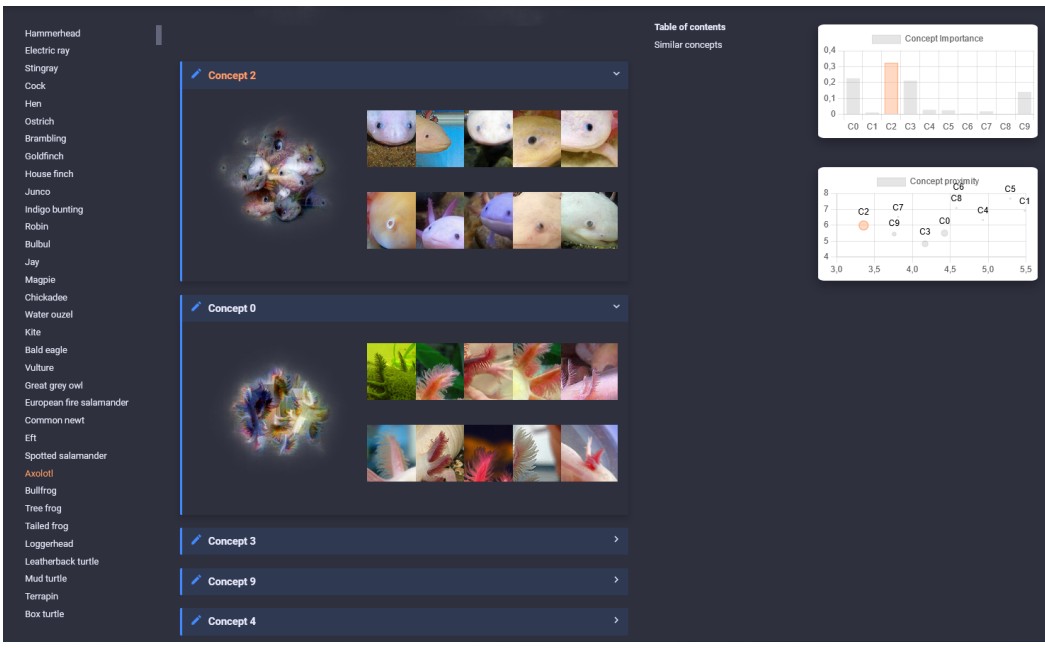

Figure S8: **Most representative crops.** If we scroll down, we get access to the crops that represent the most each of the concepts alongside the visualizations. By representing the concepts through two different approaches side by side, we reduce the effect of confirmation bias.

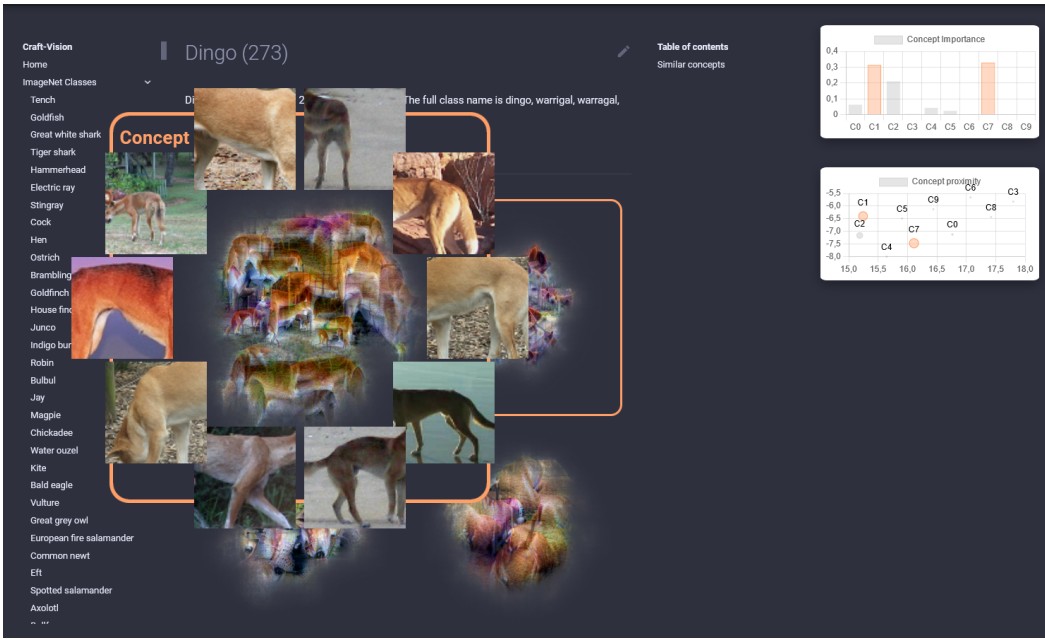

Figure S9: **Revealing the crops.** On the first overview, it is also possible to click on each concept to reveal the most representative crops.

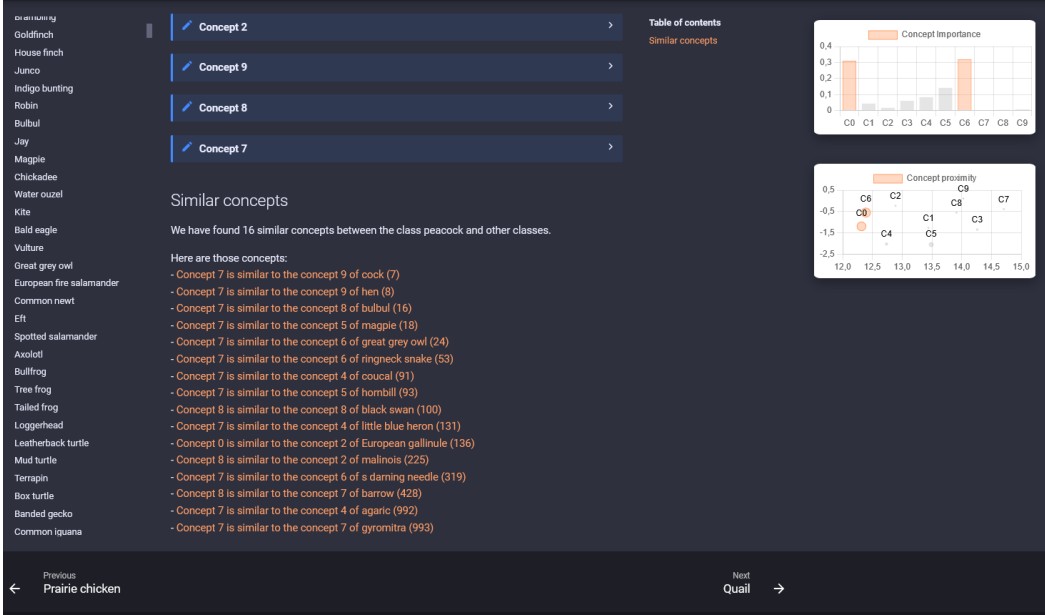

Figure S10: **Inter-class concept similarity.** Scrolling further down, we also present the concepts from other classes that are the most similar to those of the current class. This can help better understand erroneous predictions, as the model may sometimes leverage similar concepts of different classes to classify.

## C    Human psychophysical study

To evaluate MACO 's ability to improve humans' causal understanding of a CNN's activations, we conducted a psychophysical study closely following the paradigm introduced in [67]. In this paradigm, participants are asked to predict which of two query inputs would be favored by the model (i.e., maximally activate a given unit), based on example "favorite" inputs serving as a reference (i.e., feature visualizations for that unit). The two queries are based on the same natural image, but differ in the location of an occludor which hides part of the image from the model.

**Participants.**    We recruited a total of 191 participants for our online psychophysics study using Prolific (www.prolific.com) [September 2023]. As compensation for their time (roughly 7 minutes), participants were paid 1.4$. Of those who chose to disclose their age, the average age was 39 years old ($SD = 13$). Ninety participants were men, 86 women, 8 non-binary and 7 chose not to disclose their gender. The data of 17 participants was excluded from further analyses because they performed significantly below chance ($p < .05$, one-tailed).

**Design.**    Participants were randomly assigned to one of four Visualization conditions: Olah [1], MACO with mask, MACO without mask, or a control condition in which no visualizations were provided. Furthermore, we varied Network (VGG16, ResNet50, ViT) as a within-subjects variable. The specific units whose features to visualize were taken from the output layer, meaning they represented concrete classes. The classes were: Nile crocodile, peacock, Kerry Blue Terrier, Giant Schnauzer, Bernese Mountain Dog, ground beetle, ringlet, llama, apiary, cowboy boot, slip-on shoe, mask, computer mouse, muzzle, obelisk, ruler, hot dog, broccoli, and mushroom. For every class, we included three natural images to serve as the source image for the query pairs. This way, a single participant would see all 19 classes crossed with all 3 networks, without seeing the same natural image more than once (which image was presented for which network was randomized across participants). The main experiment thus consisted of 57 trials, with a fully randomized trial order.

**Stimuli.**    The stimuli for this study included 171 ((4-1)x3x19) reference stimuli, each displaying a 2x2 grid of feature visualizations, generated using the respective visualization method. The query pairs were created from each of the 57 (19x3) source images by placing a square occludor on them.

In one member of the pair, the occludor was placed such that it minimized the activation of the unit. In the other member of the pair, the occludor was placed on an object of a different class in the same image or a different part of the same object. Here, we deviated somewhat from the query geneation in [67], where the latter occludor was placed where it maximized the activation of the unit. However, we observed that this often resulted in the occludor being on the background, making the task trivial. Indeed, a pilot study ($N = 42$) we ran with such occludor placement showed that even the participants in the control condition were on average correct in $83\%$ of the trials.

**Task and procedure.** The protocol was approved by the University IRB and was carried out in accordance with the provisions of the World Medical Association Declaration of Helsinki. Participants were redirected to our online study through Prolific and first saw a page explaining the general purpose and procedure of the study (Fig. S11). Next, they were presented with a form outlining their rights as a participant and actively had to click "I agree" in order to give their consent. More detailed instructions were given on the next page (Fig. S12, Fig. S13). Participants were instructed to answer the following question on every trial: "Which of the two query images is more favored by the machine?". The two query images were presented on the right-hand side of the screen. The feature visualizations were displayed on the left-hand side of the screen (Fig. S14). In the control condition, the left-hand side remained blank (Fig. S15). Participants could make their response by clicking on the radio button below the respective query image. They first completed a practice phase, consisting of six trials covering two additional classes, before moving on to the main experiment. For the practice trials, they received feedback in the form of a green (red) frame appearing around their selected query image if they were correct (incorrect). No such feedback was given during the main experiment.

**Analyses and results.** We analyzed the data through a logistic mixed-effects regression analysis, with trial accuracy (1 vs. 0) as the dependent variable. The random-effects structure included a by-participant random intercept and by-class random intercept. We compared two regression models, both of which had Visualization and Network as a fixed effect, but only one also fitted an interaction term between the two. Based on the Akaike Information Criterion (AIC), the former, less complex model was selected ($AIC = 11481 vs. 11482$). Using this model, we then analyzed all pairwise contrasts between the levels of the Visualization variable. We found that the logodds of choosing the correct query were overall significantly higher in both **MACO** conditions compared to the control condition: $\beta_{\text{MACO } Mask} - \beta_{Control} = 0.69, SE = 0.13, z = 5.38, p < .0001; \beta_{\text{MACO } NoMask} - \beta_{Control} = 0.92, SE = 0.13, z = 7.07, p < .0001$. Moreover, **MACO** visualizations helped more than Olah visualizations: $\beta_{\text{MACO } Mask} - \beta_{Olah} = 0.43, SE = 0.13, z = 3.31, p = .005; \beta_{\text{MACO } NoMask} - \beta_{Olah} = 0.66, SE = 0.13, z = 4.99, p < .0001$. No other contrasts were statistically significant (at a level of $p < .05$). $P$-values were adjusted for multiple comparisons with the Tukey method. Finally, we also examined the pairwise contrasts for the Network variable. We found that ViT was the hardest model to interpret overall: $\beta_{ResNet50} - \beta_{ViT} = 0.49, SE = 0.06, z = 8.65, p < .0001; \beta_{VGG16} - \beta_{ViT} = 0.35, SE = 0.06, z = 6.38, p < .0001$. There was only marginally significant evidence that participants could better predict ResNet50's behavior in this task than VGG16: $\beta_{ResNet50} - \beta_{VGG16} = 0.13, SE = 0.06, z = 2.30, p = 0.056$.

Taken together, these results suggest that **MACO** indeed helps humans causally understand a CNN's activations and that it outperforms Olah's method [1] on this criterion.

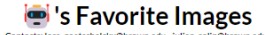's Favorite Images

Contacts: lore_goetschalckx@brown.edu - julien_colin@brown.edu

**Welcome**

This is a computational neuroscience study looking to discover how artificial intelligence (AI) "thinks". Specifically, we are interested in machines trained to recognize images. You will be asked to complete a 6-9 min task. The task consists of several trials. On each trial, you will be shown images and asked to respond by clicking the mouse.

The question you will have to respond to is:

**Which of the two images presented to you is more favored by the machine?**

The machines we are considering in this study are trained to recognize images. Participants are randomly assigned to different groups. Depending on the group you are in, you might see examples of a couple of the machine's favorite images to help you with your task. These examples are computer-generated and might look a little funky.

More detailed instructions will follow, but first please take notice of the prerequisites below.

**Prerequisites**

1. You are working on a desktop or laptop, not on a phone or a tablet.
2. You have normal or corrected-to-normal vision.

Please make a participation code using the first letter of your mother's name, the first letter of your father's name, first letter of your favorite hobby, and the last digit of your house number (e.g., PCB0). Please enter your unique code before clicking Continue.

On the next page, we will first ask you for your consent to participate in this study.

[ Continue ]

Figure S11: **Welcome page.** This is a screenshot of the first page participants saw when entering our online psychophysics study.

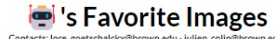

## 's Favorite Images

Contacts: lore_goetschalckx@brown.edu - julien_colin@brown.edu

**Instructions**

You are in the group who will see examples of a machine's favorite images.

On each trial, the screen will be divided into two sections:

    - LEFT: A group of example images. These are the favorite images of a machine. Usually, these images share at least one common aspect.

    - RIGHT: Two query images. In each query image, a square has been placed to cover part of the image. This part is hidden to the machine.

The question you have to answer is always the following: **Which of the two query images is more favored by the machine?** In other words, in which image do you still see the common aspect of the favorite images?

Answer by clicking on the button below the image that you believe to be favored by the machine. A grey frame will appear around your chosen image.

The image below shows you what a single trial will look like.

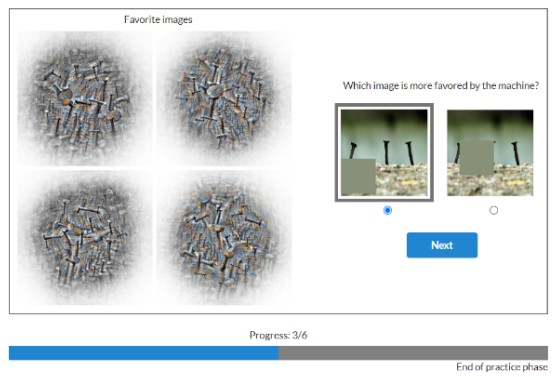

Before starting the main task, there will be several practice trials such that you can familiarize yourself with the task.

Continue

Figure S12: **Instructions page.** After providing informed consent, participants in our online psychophysics task received more detailed instructions, as shown here.

### Instructions

You are in the group who won't see examples of a machine's favorite images. Don't worry. That's not bad. We're merely interested in your sincere responses to the task.

On each trial, the screen will be divided into two sections:

 - LEFT: There won't be any images shown on this side.

 - RIGHT: Two query images. In each query image, a square has been placed to cover part of the image. This part is hidden to the machine.

The question you have to answer is always the following: **Which of the two query images is more favored by the machine?**

Answer by clicking on the button below the image that you believe to be favored by the machine. A grey frame will appear around your chosen image.

The image below shows you what a single trial will look like.

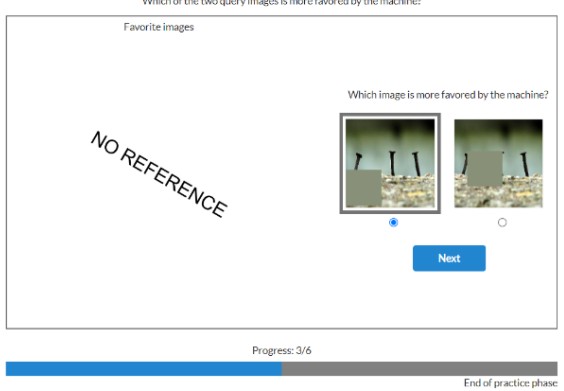

Before starting the main task, there will be several practice trials such that you can familiarize yourself with the task.

Continue

Figure S13: **Instructions page for control condition.** After providing informed consent, participants in our online psychophysics task received more detailed instructions, as shown here. If they were randomly assigned to the control condition, they were informed that they would not see examples of the machine's favorite images.

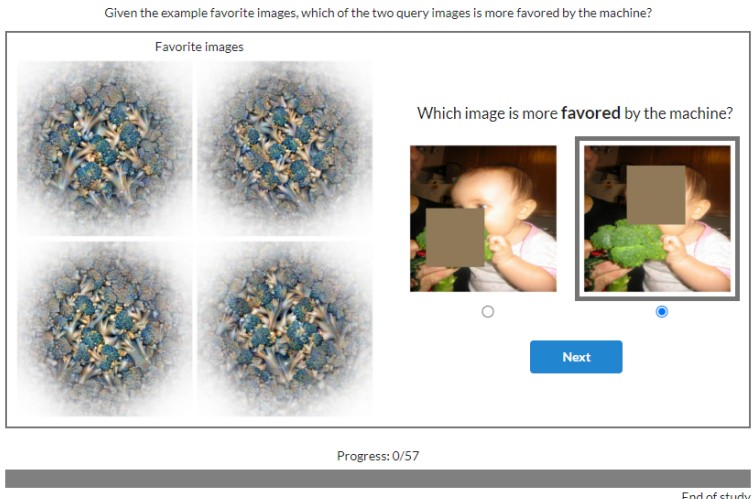

Figure S14: **Example trial.** On every trial of our psychophysics study, participants were asked to select which of two query images would be favored by the machine. They were shown examples of the machine's favorite inputs (i.e., feature visualizations) on the left side of the screen.

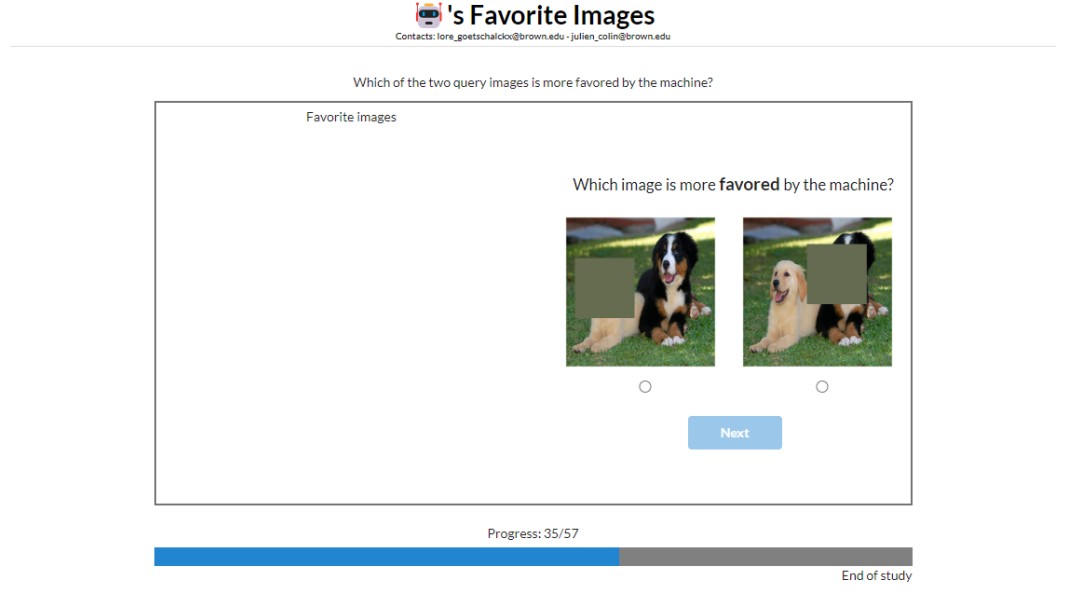

Figure S15: **Example trial in the control condition.** On every trial of our psychophysics study, participants were asked to select which of two query images would be favored by the machine. In the control condition, they were not shown examples of the machine's favorite inputs and the left side of the screen remained empty.