# OpenReview forum: "Unlocking Feature Visualization for Deep Network with MAgnitude Constrained Optimization"
_NeurIPS.cc/2023/Conference — NeurIPS 2023 poster_

### Official Review · Reviewer_mBBq · 2023-06-15

**Soundness:** 3 good
**Presentation:** 3 good
**Contribution:** 4 excellent
**Rating:** 7
**Confidence:** 5

**Summary:**

The paper proposes MACO, a feature visualization technique that optimizes image phase but keeps the magnitude of the Fourier spectrum constant. This method is similar, in spirit, to the 'classic' feature visualizations by Olah et al. (2017), while at the same time enabling feature visualizations for architectures where the classic approach previously struggled with (such as vision transformers or other non-VGG architectures). The paper lists three core contributions:

1. MACO, the new feature visualization technique,
2. Combining feature visualization with feature attribution,
3. Quantitative metrics to measure the success of the proposed method

Overall, (1.) can be seen as the most significant contribution whereas (2.) is an application and (3.) is a supporting contribution.

**Strengths:**

1. The paper is very well written and a pleasure to read
2. The proposed approach, MACO, is refreshingly simple (e.g., fewer transformations needed to make it work) and has a clear motivation from human perception (importance of phase spectrum for human perception).
3. Making feature visualizations work for a broader range of architectures is a desirable goal, and there hasn't been much progress in this direction over the last few years
4. The authors make the findings explorable via a website (currently screenshots in the Appendix)
5. The paper contains a quantitative evaluation for at least some findings. The 'plausibility score' and the 'FID score' both make a very big assumption (natural images = best, which may not be the case for the model: in fact, in the introduction MACO is proposed exactly since the feature visualizations based on generative models like GANs make this assumption). Nevertheless, in a field that's predominantly evaluated 'by eye' (nice-looking visualizations) it's great that they perform at least some quantitative evaluations.

**Weaknesses:**

1. The strongest contribution is MACO itself. The approach of combining attribution and feature visualization, on the other hand, sounds like a great idea if at least one of those methods, and ideally both, would be reliable. In particular, many attribution methods are deeply problematic [1,2]. The particular method employed in the paper is similar to SmoothGrad which often produces attributions that are as reliable as random guessing [2]. I very much hope that the approach of combining attribution and feature visualization can be backed up by quantitative, falsifiable evaluation. If it cannot, I would be inclined to question its inclusion in the paper.

2. The paper makes the big assumption that feature visualizations which resemble natural images are better (explained above), which is a prior that's independent of the model and can thus lead to misleading visualizations if the assumption isn't met by the investigated model.

3. The claim that "we supply the field of XAI with a reliable tool for explaining the semantic ('what' information) of modern vision models" sounds a bit too strong in my ears. Given the well-known limitations of feature visualizations (e.g., hard to falsify [3], hard to quantify, unreliable [very recent work 4]), I think it might be worth formulating this a bit more carefully, for instance as "we supply the field of XAI with a tool for exploring hypotheses about the semantic ('what' information) of modern vision models".

[1] Zhou et al. https://ojs.aaai.org/index.php/AAAI/article/view/21196/20945

[2] Bilodeau et al. https://arxiv.org/pdf/2212.11870.pdf

[3] Leavitt & Morcos https://arxiv.org/pdf/2010.12016.pdf

[4] Geirhos et al. https://arxiv.org/pdf/2306.04719.pdf


**Questions:**

1. How can the feature attribution + feature visualization aspect be quantified and falsified? (see above)

2. For every single feature visualization figure shown in the paper and the supplementary materials, how were the units chosen? If at random, great; if not I would encourage the authors to either update those figures with randomly chosen classes (and channels etc.) or to state, in the figure caption, that they were not chosen randomly (a.k.a. cherry-picked).

3. Are you planning to open-source the technique & code, e.g. as a standalone library / contribution to an existing library to facilitate using the method? (Not required of course, but might be something worth thinking about)

I am willing to increase my score if my concerns can be addressed; in particular regarding feature attribution + feature visualization, random vs. cherry-picked visualizations, and toning down the claim a bit.

Minor:

3. In line 37, it would be useful to define 'faithful'

4. In lines 178-180, it would be useful to quantify this statement, e.g. in the appendix

**Limitations:**

Appendix A contains a great discussion on potential limitations, including the fact that "For a feature visualization to be truly informative about the model (e.g., about those spurious features), it might need to generate visualizations that stray away from the spectrum of natural images, and thus, yield worse scores using our metrics."

---

> ### Author Rebuttal · Authors · 2023-08-08
>
> We thank the reviewer for their time and for providing this detailed review with very relevant comments.
>
> ## Weaknesses:
>
> **The strongest contribution is MACO itself. The approach of combining attribution and feature visualization, on the other hand, sounds like a great idea if at least one of those methods, and ideally both, would be reliable. In particular, many attribution methods are deeply problematic [1,2]. The particular method employed in the paper is similar to SmoothGrad which often produces attributions that are as reliable as random guessing [2]. I very much hope that the approach of combining attribution and feature visualization can be backed up by quantitative, falsifiable evaluation. If it cannot, I would be inclined to question its inclusion in the paper.**
>
> The only overlap with SmoothGrad is in that we use an average of gradients. However, unlike SmoothGrad, we do not use gradients with respect to sampled noise, but rather use the gradients obtained at each MACO’s optimization steps to compute the transparency masks. Hence, the fact that “SmoothGrad [...] often produces attributions that are as reliable as random guessing [2]” does not apply for us. That said, we agree that the usefulness of using SmoothGrad might be further assessed with psychophysics experiments that are however beyond the scope of the paper (as an example, [47, 48] are fully dedicated to such an evaluation, showing that it is a whole research work on its own).
>
> [1] "What I Cannot Predict, I Do Not Understand: A Human-Centered Evaluation Framework for Explainability Methods," Colin et al., NeurIPS 2022.
>
> **The paper makes the big assumption that feature visualizations which resemble natural images are better (explained above), which is a prior that's independent of the model and can thus lead to misleading visualizations if the assumption isn't met by the investigated model.**
>
> We apply the “naturalness” prior in the Fourier space, which does not mean that we obtain a naturalness prior in the pixel space. Looking at the illustrations (on the paper and website), one can see that images do not seem so natural. We would rather say that we impose a human-interpretability prior, which seems legitimate since feature visualizations are designed to be interpretable by humans. Still, note that to limit biases that would lead to visualizations that are too far from the model's typical features, we report the plausibility (with the OOD score) as a key metric to assess the feature visualization.
>
> **The claim that "we supply the field of XAI with a reliable tool for explaining the semantic ('what' information) of modern vision models" sounds a bit too strong in my ears. Given the well-known limitations of feature visualizations (e.g., hard to falsify [3], hard to quantify, unreliable [very recent work 4]), I think it might be worth formulating this a bit more carefully, for instance as "we supply the field of XAI with a tool for exploring hypotheses about the semantic ('what' information) of modern vision models".**
>
> We have edited this sentence, and toned down this and other claims– especially considering recent articles that highlight issues with these methods in a general sense. We thank the reviewer for this suggestion.
>
> ## Questions
>
> **For every single feature visualization figure shown in the paper and the supplementary materials, how were the units chosen? If at random, great; if not I would encourage the authors to either update those figures with randomly chosen classes (and channels etc.) or to state, in the figure caption, that they were not chosen randomly (a.k.a. cherry-picked).**
>
> **Are you planning to open-source the technique & code, e.g. as a standalone library / contribution to an existing library to facilitate using the method? (Not required of course, but might be something worth thinking about)**
>
> For the sake of transparency, we release the website that showcases visualizations of concepts for each of the 1000 imagenet’s classes. We encourage the reviewer  to explore the anonymous site we provided (https://anon1285.github.io/neurips1285/).
>
> Regarding the code, it is ready and will be released and available in two ways: 1) in a standalone repository, and 2) integrated into an XAI library.
>
> ## Limitations
>
> **Appendix A contains a great discussion on potential limitations, including the fact that "For a feature visualization to be truly informative about the model (e.g., about those spurious features), it might need to generate visualizations that stray away from the spectrum of natural images, and thus, yield worse scores using our metrics."**
>
> Thank you very much. If the paper is accepted, we would like to use the supplementary page to incorporate the limitations directly into the main text.

---

> > ### Comment · Reviewer_mBBq · 2023-08-11
> > **Thanks for the reply!**
> >
> > I would like to thank the authors for their detailed and helpful reply. I have read the author's rebuttal, as well as the other reviewer's comments and I support publication at NeurIPS.
> >
> > In particular, it is worth noting that in response to the reviews, the authors have released an anonymous version of their website and it looks amazing - for instance, it makes it very easy to relate a visualized concept to natural images that strongly activate that specific concept.
> >
> > Furthermore, I appreciate that the authors provided details on how they will open-source their codebase, which will be a valuable contribution to visualize concepts e.g. for network architectures where current methods struggle.
> >
> > While I appreciate the author's explanation regarding the difference to SmoothGrad, I would still have loved to see quantitative, falsifiable evaluation of the feature attribution aspect.
> >
> > An open question I still have is the following: For every single feature visualization figure shown in the paper and the supplementary materials, how were the units chosen? If at random, great; if not I would encourage the authors to either update those figures with randomly chosen classes (and channels etc.) or to state, in the figure caption, that they were not chosen randomly (a.k.a. cherry-picked / manually selected). I very much appreciate the released website, but for the paper itself it would be helpful to state whether the units were chosen by the authors or by a random seed. Both choices are completely fine but I think readers should know.
> >
> > I am increasing my score from 6 to 7.

---

> > > ### Author Response · Authors · 2023-08-14
> > >
> > > Thank you very much for your thoughtful and supportive feedback.
> > >
> > > It's particularly gratifying to hear that you found clear connections between visualized concepts and corresponding natural images with strong activations.
> > >
> > > Regarding the selection of units for feature visualization figures, your question is valid, and we thank you for raising this point. We will ensure that the figure captions explicitly state whether units were chosen by the authors or based on a random seed, providing transparency for our readers.
> > >
> > > Once again, thank you for revising your score and for your valuable support!

---

### Official Review · Reviewer_UD3m · 2023-07-02

**Soundness:** 3 good
**Presentation:** 3 good
**Contribution:** 2 fair
**Rating:** 4
**Confidence:** 3

**Summary:**

## Summary

This paper presents MACO, a simple and novel method to generate feature visualization for deep neural networks. MACO uses a constraint to optimize feature visualization in the space of natural images. It also presents an attribution-based mask to augment generated images to highlight spacial importance. To evaluate MACO, three metrics (transferability, plausibility, and alignment with natural images) are proposed. Evaluation study shows that MACO generates more useful feature visualization and is scalable to modern vision models such as ViT.

**Strengths:**

## Strengths

S1: This paper aims to improve the existing feature visualization method, which is popular in explaining vision models. This is a very critical and timely research topic.

S2: The proposed method MACO is sound, and the intuition of optimizing for image phase over magnitude is reasonable. It's good that MACO doesn't need to use any strong parametric image priors.

S3: The paper is well-written and easy to follow. I am looking forward to seeing the interactive visualization website!

**Weaknesses:**

## Major weaknesses

W1: The evaluation can be improved. The goal of feature visualization is to help human users interpret computer vision models. However, I don't see a strong connection between the proposed evaluation scores (plausibility, FID, and transferability scores) and explainability. These scores assume that *feature visualizations more similar to natural images are better at explaining ML models.* However,  this paper does not provide strong evidence that this hypothesis is true. This paper would be much stronger if it can directly highlight MACO improves explainability.

W2: Related to W1, a user study is necessary for this paper. I found some of the figures very hard to interpret. Studying how real users (e.g., ML developers and researchers) would interpret and use MACO-generated feature visualizations in their workflows would yield many interesting insights.

## Minor weaknesses

M1: The abstract can be improved by giving more details about the MACO technique. The current one-sentence, "It consists in optimizing solely an image's phase spectrum while keeping its magnitude constant to ensure that the generated explanations lie in the space of natural images." might be too abstract.

**Questions:**

1. How does MACO improve ML explainability?
2. How would an ML practitioner/researcher use MACO to improve their ML models?

**Limitations:**

Yes.

---

> ### Author Rebuttal · Authors · 2023-08-08
>
> First of all, we thank the reviewer for taking the time to read our article.
>
> ## Major
>
> **W1: The evaluation can be improved. The goal of feature visualization is to help human users interpret computer vision models. However, I don't see a strong connection between the proposed evaluation scores (plausibility, FID, and transferability scores) and explainability. These scores assume that feature visualizations more similar to natural images are better at explaining ML models. However, this paper does not provide strong evidence that this hypothesis is true. This paper would be much stronger if it can directly highlight MACO improves explainability.**
>
> Note that the scores do not only assume that visualizations more similar to natural images are better at explaining ML models. This statement is true for FID, but plausibility, based on OOD score, measures how close the visualizations are to the features that the model has constructed during the training, whether natural or not. As for transferability, this score ensures that the visualization is not excessively adversarial. As emphasized in the general comment, the proposed metrics are to be considered together, not independently.
> Note that the naturalness of feature visualization is known to be highly correlated with their interpretability [47].
>
> Taking a look at our website (e.g. https://anon1285.github.io/neurips1285/classes/espresso/), it is clear that feature visualization is complementary to concept-based techniques [29, 46], which illustrates how it can improve explainability.
>
> **W2: Related to W1, a user study is necessary for this paper. I found some of the figures very hard to interpret. Studying how real users (e.g., ML developers and researchers) would interpret and use MACO-generated feature visualizations in their workflows would yield many interesting insights.**
>
> One interesting follow up for future work would indeed be to evaluate those metrics using human psychophysics. Many reviewers have suggested this idea, and we acknowledge the necessity of such an experiment but we consider it beyond the scope of the present  paper. As an example, [47, 48] are fully dedicated to such an evaluation, showing that it is a whole research work on its own. Rather, as a first step this paper focuses on MACO as an improvement over existing Feature Visualization techniques.
>
> ## Minor
>
> **M1: The abstract can be improved by giving more details about the MACO technique. The current one-sentence, "It consists in optimizing solely an image's phase spectrum while keeping its magnitude constant to ensure that the generated explanations lie in the space of natural images." might be too abstract.**
>
> We rephrased the sentence and added it to the manuscript. We thank the reviewer for the suggestion.
>
> ## Questions
>
> **1- How does MACO improve ML explainability?**
>
> **2- How would an ML practitioner/researcher use MACO to improve their ML models?**
>
> MACO is a Feature Visualization technique, so its use in ML explainability for practitioners would be the same as other classical Feature Visualization techniques (finding spurious correlation, understanding the inner workings of the model,...). Importantly, it might not be sufficient to fully explain a model, and rather has to be taken as a tool towards making models more interpretable. A good example of how it can be used in combination with other XAI tools is the website that showcases its complementarity with concept based explanations ( https://anon1285.github.io/neurips1285/).

---

> > ### Comment · Reviewer_UD3m · 2023-08-13
> > **Thank you for the detailed reply!**
> >
> > Thank you for the detailed reply! The interactive website looks neat.
> >
> > This work focuses on feature **visualization** and ML **explainability**, where both topics **directly target** human users (i.e., viewers of the visualizations and people being explained to). The evaluation is less convincing without a user study. I still think a user study evaluation is necessary and within the scope of the paper.

---

> > > ### Author Response · Authors · 2023-08-14
> > >
> > > We greatly appreciate your thoughtful evaluation and recognition of the value our website brings.
> > >
> > > We acknowledge your point about the potential benefits of a user study in understanding MACO's impact on ML explainability. While we recognize this aspect, we'd like to emphasize that our paper introduces the first method allowing to effectively visualize deep models and we believe that MACO has the potential to greatly benefit the research community by enabling the exploration and research that were previously infeasible for modern networks.
> > >
> > > Furthermore, our website currently serves as a valuable resource for identifying misleading correlations and gaining insights into the nuances of ImageNet classes. This tangible utility underscores MACO's practical advantages, even without an extensive user study.
> > >
> > > We kindly request a reconsideration of our paper's evaluation, taking into account its innovative nature, scalability, and the practical value our website contributes to enhancing model transparency.
> > >
> > > Thank you for your thoughtful consideration.

---

### Official Review · Reviewer_qNv3 · 2023-07-05

**Soundness:** 2 fair
**Presentation:** 3 good
**Contribution:** 2 fair
**Rating:** 4
**Confidence:** 5

**Summary:**

This paper proposes a small but apparently effective modification to the common technique of feature visualizations to improve the visual quality of the produced explanations. Specifically, this approach uses a different image parameterization producing images with power spectra similar to that of natural/dataset images.

**Strengths:**

- The modification to the feature visualization procedure is well-motivated and described. The method is easy to understand.

**Weaknesses:**

- Section 4: The authors claim that they are the first to introduce quantitative methods for comparing feature visualizations. This is, however, incorrect as Borowski et al. [47] and Zimmermann et al. [48] already introduce evaluation schemes for measuring the helpfulness of visualizations.
- This study does not include any psychophysical evaluation/study investigating the helpfulness of the produced visualizations for humans. This is, however, a crucial part when developing new XAI methods as was noted by different earlier studies. I encourage the authors to test their method using the quantitative psychophysical benchmark introduced by Borowski et al. 2020, as this will drastically increase my confidence in this work and, hence, also my rating.
- While the authors list an interactive website as one of their contributions, we sadly cannot access and test it. While I understand that granting us access to this without risking a violation of the double-blind peer review process is difficult, I'm feeling uncomfortable giving the paper a great rating (based on this claimed contribution) without being able to review this website, too.

**Questions:**

- L161: The transformations used by Olah et al. can also be summarized as just a few, namely random cropping with zoom and rotations.
- L179: Showing a visual comparison would be appreciated.
- L202ff: Motivating why these augmentations are used/needed would be great. This sounds somehow contracting the earlier paragraphs where the authors say they want to remove these augmentations.
- L205f: This is a bit misleading; Olah et al. also didn't use explicit regularizers.
- L212: Unclear what is meant by confirmation bias in this context and how the proposed method addresses this.
- L216ff: Why is such a visualization more helpful/better for a user than the unmasked one? Performing a user study for this could be helpful.
- L233f: Using the plausibility score to judge a visualization appears to assume that the classifier is doing a great job and learns something that is perfectly aligned with how humans perform this classification.

**Limitations:**

- The authors do not discuss any limitations in the main part of their paper.

---

> ### Author Rebuttal · Authors · 2023-08-08
>
> We thank the reviewer for taking the time to read our article, and we sincerely appreciate their review.
>
> - **Section 4: The authors claim that they are the first to introduce quantitative methods for comparing feature visualizations. This is, however, incorrect as Borowski et al. [47] and Zimmermann et al. [48] already introduce evaluation schemes for measuring the helpfulness of visualizations.**
>
> - **This study does not include any psychophysical evaluation/study investigating the helpfulness of the produced visualizations for humans. This is, however, a crucial part when developing new XAI methods as was noted by different earlier studies. I encourage the authors to test their method using the quantitative psychophysical benchmark introduced by Borowski et al. 2020, as this will drastically increase my confidence in this work and, hence, also my rating.**
>
> We agree with qNv3, that Borowski et al [47] and Zimmermann et al [48] have provided insightful contributions to evaluate feature visualization techniques. However, the metrics proposed are all based on psychophysics that require heavy experimental protocols and that need to be re-conducted on every new network one wants to evaluate. Those psychophysics experiments are hardly compatible with the high pace of new networks in the ML field. As opposed to [47] and [48] we wanted to propose metrics that are easily computable, scalable to novel networks and that could easily serve as a means of comparison for the entire XAI community. In the end, we see these metrics as complementary to psychophysics-based evaluations that are used for determining the helpfulness of feature visualizations to humans. This is why conducting such an experiment is one avenue for future work, but we consider it to be beyond the scope of this paper, that is focused on MACO as an improvement of existing Feature Visualization techniques.
>
> - **While the authors list an interactive website as one of their contributions, we sadly cannot access and test it. While I understand that granting us access to this without risking a violation of the double-blind peer review process is difficult, I'm feeling uncomfortable giving the paper a great rating (based on this claimed contribution) without being able to review this website, too.**
>
> We have updated an anonymized version here: https://anon1285.github.io/neurips1285/classes/espresso/. You can browse all 1000 imagenet’s classes using the search bar at the top or the menu at the left.
>
> - **L161: The transformations used by Olah et al. can also be summarized as just a few, namely random cropping with zoom and rotations.**
>
> The official implementation of Olah et. al [1] utilizes a cascade of 10 augmentations, whereas we only use 2.
>
> https://github.com/tensorflow/lucid/blob/6dcc927e4ff4e7ef4d9c54d27b0352849dadd1bb/lucid/recipes/activation_atlas/render.py#L130
>
> - **L179: Showing a visual comparison would be appreciated.**
>
> A visual comparison is already main in Figure 1 of the main manuscript (Fourier FV). We will clarify upon acceptance that Fourier FV refers to the methods of Olah et al.
>
> - **L202ff: Motivating why these augmentations are used/needed would be great. This sounds somehow contracting the earlier paragraphs where the authors say they want to remove these augmentations.**
>
> The simplicity of our method allows us to drastically reduce the number of augmentations (from 10 to 2), and therefore the number of hyperparameters, but we still need those two augmentations to stabilize the optimization procedure.
>
> - **L205f: This is a bit misleading; Olah et al. also didn't use explicit regularizers.**
>
> It is true that although [1] uses many regularizations, those might not be considered explicit. We removed the word “explicit” to avoid confusion.
>
> - **L212: Unclear what is meant by confirmation bias in this context and how the proposed method addresses this**
> - **L216ff: Why is such a visualization more helpful/better for a user than the unmasked one? Performing a user study for this could be helpful.**
>
> (L212 - L216) Without our method, “Visualizations often suffer from repeated patterns or unimportant elements in the generated images.”. In that case, the confirmation bias would consist in interpreting unimportant elements as meaningful explanations. Our methods alleviates this problem by masking unimportant elements.
>
> - **L233f: Using the plausibility score to judge a visualization appears to assume that the classifier is doing a great job and learns something that is perfectly aligned with how humans perform this classification.**
>
> Apologies for any confusion caused by the naming of the metric. It is not about plausibility with respect to humans but rather with respect to the model (OOD). This metric quantifies the extent to which the internal feature paths are conventional (plausible) for maximizing the objective (see the motivation described in the general message, point 1.).
>
> ## Limitations
>
> - **The authors do not discuss any limitations in the main part of their paper.**
>
> Appendix A is entirely dedicated to a thorough discussion of the limitations. We did not include it in the main text because of the page limit but we plan to do so upon acceptance – thanks to the extra page.

---

> > ### Comment · Reviewer_qNv3 · 2023-08-13
> >
> > Thank you for your responses, I will reply to open points in chronological order.
> >
> > **Section 4: The authors claim that they are the first to introduce quantitative methods for comparing feature visualizations. This is, however, incorrect as Borowski et al. [47] and Zimmermann et al. [48] already introduce evaluation schemes for measuring the helpfulness of visualizations.**
> >
> > While I understand your point that the existing evaluation metrics are time-consuming to compute, as they require human participants, I would like to point out that they achieve something you cannot automate: namely, they establish a clear connection between any explanation method and how useful it is for a _human_ to understand a neural network. Automated metrics like the ones you propose here cannot replace this and can only be seen as a first evaluation step in the sense that if a new method fails in your metric it is unlikely to be helpful for humans, but methods that perform well in your metric can still be unhelpful for humans. I believe this is not a fringe opinion but rather represents a common belief in the current community. Therefore, I'm a bit hesitant to recommend acceptance of this paper when the paper suggests these automated metrics are (mostly) all you need.
> >
> > **L161: The transformations used by Olah et al. can also be summarized as just a few, namely random cropping with zoom and rotations.**
> > I don't see how the reference line invalidates my point: while their code calls multiple augmentations, they are mostly identical with different hyperparameters and can be - in spirit - reduced to random crops/zooms/rotations. Also, note that [other parts of their code](https://github.com/tensorflow/lucid/blob/master/lucid/optvis/transform.py#L165), used for most feature visualizations, are substantially simpler.
> >
> > **L179: Showing a visual comparison would be appreciated.**
> > Given the importance of this comparison, I would still appreciate it if you can include a bigger version of this figure with more details in the next version of the manuscript to give readers a better understanding of how these visualizations differ. Also, on a related note, it might be interesting to see how your different design choices impact the final visualization compared to those of Olah et al.
> >
> > **L216ff: Why is such a visualization more helpful/better for a user than the unmasked one? Performing a user study for this could be helpful.**
> > Thank you for this explanation. Given the criticism the field of saliency methods has seen in the past years and that your method of masking out image elements is similar to those saliency methods, I would find a discussion of the problems and - as mentioned above - a user study to truly evaluate this helpful to trust your method.

---

> > > ### Author Response · Authors · 2023-08-14
> > >
> > > We thank the reviewer for their feedback.
> > >
> > > > "I'm feeling uncomfortable giving the paper a great rating (based on this claimed contribution) without being able to review this website"
> > >
> > > In addressing your inquiry regarding  the website, we have implemented measures to ensure its anonymity, thereby enabling your access while upholding the integrity of the double-blind peer review process and we hope you will explore it if you have not yet.
> > >
> > > https://anon1285.github.io/neurips1285/
> > >
> > > > while their code calls multiple augmentations, they are mostly identical with different hyperparameters and can be - in spirit - reduced to random crops/zooms/rotations.
> > >
> > > In addressing your inquiry regarding Olah et al. transformations, we have provided a specific reference to their implementation that highlights our method's distinction in terms of the number of augmentations used. We believe our clarification adequately addresses this issue.
> > >
> > > > Given the importance of this comparison, I would still appreciate it if you can include a bigger version of this figure with more details
> > >
> > > In addressing your inquiry for a visual comparison, we have enhanced Figure 1 of the main manuscript to provide a more comprehensive illustration of the disparities between our visualizations and those of Olah et al.
> > >
> > > ---
> > > We believe we have addressed your concerns with our rebuttal and new results. We would like to emphasize that we are presenting the first feature visualization method capable of scaling to deep networks, generating non-adversarial feature visualizations on modern
> > >  architectures.
> > >
> > > We share your viewpoint on the significance of human experiments and acknowledge the merit of Zimmerman's great contributions.
> > > We are actively considering the integration of these aspects in an extended version of our paper. But, we believe our work already have major contributions (as noted by some reviewers). Your consideration of these aspects when evaluating our paper would be highly appreciated.

---

### Official Review · Reviewer_6q7R · 2023-07-08

**Soundness:** 2 fair
**Presentation:** 3 good
**Contribution:** 2 fair
**Rating:** 4
**Confidence:** 4

**Summary:**

The paper proposes MACO, a feature visualization method which focuses the generation process of the visualizations on the image's phase spectrum while keeping the magnitude constant. This makes the generated explanation visualizations lie in the space of natural images, and hence, clearer for visual inspection.

In addition three metrics are proposed, i.e. plausability, FID and transferability, aimed at the quantitative comparison of the visualizations produced by feature visualization methods.

The proposed method is evaluated in several architectures, e.g. FlexiViT, ViT, ResNetV2, and compared against related methods that do not rely on image priors.

**Strengths:**

- Presentation of the manuscript is good, some of the design choices are properly motivated.
- The proposed method is validated in a variety of architectures.
- The propose method effectively removes the need for hyperparameters and explicit regularization.

**Weaknesses:**

- W1: The proposed method for generating the explanation visualizations raises from the observation that human recognition of objects in images is driven not by magnitude but by phase (l.190). By following this principled aren't we assuming that the model being explained operates in the same manner as humans? aren't we risking hiding features potentially useful by the model and not necessarily following human-related processes?

- W2: The manuscript proposes three metrics/scores (plausibility, FID and transferability) to quantitatively compare the feature visualizations produced by different methods.

Plausability and FID, are related to the quantification of the similarly between the distribution of a given visualization (or set of visualizations) from a given class, and the distribution of natural images from that class. While the notion behind these two metrics is clear, especially for FID, I am not sure how sensitive these two metrics might be to trivial inputs, e.g. an average image of the class of interest? I would expect this input to get a very good value on the two. Perhaps a similar trend might be observed for methods that exploit image priors.

transferability, is related to the activations/prediction triggered by the visualizations under evaluation when fed to other models trained on the same datasets. In this regard, I fail to see how following this procedure will help 1) assess the validity of the visualization and 2) its relationship with respect to the model it was extracted from.

In l.262 it is claimed that the three proposed scores provide a more complete and accurate evaluation of the feature visualization methods, based on the observations made above, this statement might be inaccurate

- W3: Besides the quantitative evaluation using the three proposed metrics, with results reported in Table 1 and Table 2, all the other sources of validation are purely qualitative.


**Questions:**

- Please see W1
- What would the Plausability and FID metrics report on average images and methods that exploit image priors?
- What is the main motivation behind the transferability metric? In the way it is defined, how would it help assess the validity of the visualizations and its relationship with respect to the model it was extracted from?

**Limitations:**

- Visualizations somewhat biased towards human perception/recognition principles (risk of low fidelity to the model being interpreted)
- While a good first attempt and step on the right direction, the proposed quantitative metrics might not be the most optimal.
- Good part of the evaluation is qualitative.

---

> ### Author Rebuttal · Authors · 2023-08-08
>
> We thank the reviewer for taking the time to read our article and for the valuable feedback.
>
>  ## Weaknesses
>
> - **W1: The proposed method for generating the explanation visualizations raises from the observation that human recognition of objects in images is driven not by magnitude but by phase (l.190). By following this principle aren't we assuming that the model being explained operates in the same manner as humans? aren't we risking hiding features potentially useful by the model and not necessarily following human-related processes?**
>
> Regarding the optimization phase, the reviewer is absolutely right, and we have acknowledged this limitation in the limitations section of Appendix A. However, the end goal of feature visualization is to provide human-interpretable explanations. If frequencies that are not visible to the human eye are present in the feature visualization, they will not be conveyed and comprehended by humans. MACO proposes to maximize the objective without assuming that it operates like humans – as evidenced by the visualizations that are far from looking like natural images–while adhering to understandable features for humans. In summary, avoiding unrealistic frequencies is both a limitation and a feature of the approach. In addition to this limitation, our method is less constrained than generative-based feature visualization methods for which it is impossible to untangle the true contributions of the model under study from those of the generative prior model [1].
>
> - **W2: The manuscript proposes three metrics/scores (plausibility, FID and transferability) to quantitatively compare the feature visualizations produced by different methods.**
>
> **Plausability and FID, are related to the quantification of the similarity between the distribution of a given visualization (or set of visualizations) from a given class, and the distribution of natural images from that class. While the notion behind these two metrics is clear, especially for FID, I am not sure how sensitive these two metrics might be to trivial inputs, e.g. an average image of the class of interest? I would expect this input to get a very good value on the two. Perhaps a similar trend might be observed for methods that exploit image priors.**
>
> **transferability, is related to the activations/prediction triggered by the visualizations under evaluation when fed to other models trained on the same datasets. In this regard, I fail to see how following this procedure will help 1) assess the validity of the visualization and 2) its relationship with respect to the model it was extracted from.**
>
> **In l.262 it is claimed that the three proposed scores provide a more complete and accurate evaluation of the feature visualization methods, based on the observations made above, this statement might be inaccurate**
>
> Regarding the metrics, we have provided a detailed response in our general message (point 1). In summary, we agree with the reviewer’s point that each metric, taken independently, may not be perfect, but together, they form a coherent whole.
>
> We tested the reviewer’s  idea of assessing the metric for the average of each class on 30 random classes, and it turns out that, concerning the reviewer’s example, the baseline score for Plausibility is bad (1977 for ResNet) and the transferability score is at zero.
>
> - **W3: Besides the quantitative evaluation using the three proposed metrics, with results reported in Table 1 and Table 2, all the other sources of validation are purely qualitative.**
>
> The field of Feature Visualization suffers from this drawback that there is no accessible, reproducible quantitative metrics available. Previous contributions – accepted by the community – only relied on qualitative assessment [1,16,19,20], or quantitative based on psychophysical, hard-to-reproduce experiments [47,48]. We agree that it is a serious drawback, and that is why we introduce quantitative metrics as a first step towards a more rigorous and accessible assessment. On the other hand, explainability always involves human interpretability, and that is why qualitative assessment is as important as quantitative assessment. Hence, we do not see the fact that “good part of the evaluation is qualitative” as a limitation, as long as it is accompanied by quantitative ones.
>
> ## Questions
>
> - **What would the Plausability and FID metrics report on average images and methods that exploit image priors?**
> - **What is the main motivation behind the transferability metric? In the way it is defined, how would it help assess the validity of the visualizations and its relationship with respect to the model it was extracted from?**
>
> The Plausibility and FID metrics would prove those images to be better than the feature visualizations generated via MaCo, but they would depend much less (or not at all) on the model under study. As for the transferability metric, its main goal is to make sure that the features highlighted by the visualizations are universal to other models (and thus, possibly to humans as well).
>
> ## Limitations
>
> - **Visualizations somewhat biased towards human perception/recognition principles (risk of low fidelity to the model being interpreted)**
> - **While a good first attempt and step on the right direction, the proposed quantitative metrics might not be the most optimal.**
> - **Good part of the evaluation is qualitative**
>
> As for the limitations, we invite the reviewer to take a look at our limitations section in Appendix A, that we will also move into the main paper upon acceptance.

---

> > ### Comment · Reviewer_6q7R · 2023-08-14
> >
> > I thank the feedback provided to my review.
> > To a good extent, the provided rebuttal further elaborated and clarified some of my initial concerns.
> >
> > I agree on that the line of work around Feature Visualization does suffer from the major drawback of a lack of a protocol for the quantitative evaluation of their methods. In this regard, in the manuscript is stated that the three proposed metrics (Plausibility, FID, and Transferability) on their own might not be sufficient, but together they form a coherent whole. May you elaborate on how these three metrics together produce the properties that are desirable for an explanation method?

---

> > > ### Author Response · Authors · 2023-08-15
> > >
> > > Thank you for your time and reponse!
> > >
> > > Our objective in combining Plausibility, FID, and Transferability metrics is to ensure that the generated images not only achieve high logit scores but also exhibit a wide and rich range of features in line with the distribution. This guarantees that the generated visuals encompass a comprehensive array of potential features, contributing to a deeper understanding of the underlying model.
> > >
> > > Furthermore, by placing a strong emphasis on Plausibility and OOD, our goal is to ensure that the generated features align with the model's typical internal pathways and adhere to the boundaries of credible internal features, avoiding any generation of explanations that might mislead or deviate from the model's "normal" behavior.
> > >
> > > In brief, an effective feature visualization method encompasses a diverse spectrum of possible features (FID), employing regular internal pathways while avoiding adversarial outcomes and out-of-distribution anomalies (Plausibility, OOD).
> > >
> > > We are confident that these metrics reflect MACO's results in producing diverse and sharp feature visualizations, as showcased on the anonymized website (https://anon1285.github.io/neurips1285/classes/fig, clicking on the feature visualization enables you to view the images that trigger this specific feature).
> > >
> > > Please let us know if you need further clarification.

---

> > > > ### Comment · Reviewer_6q7R · 2023-08-19
> > > >
> > > > Thanks for the additional details.
> > > > I will take it into account when making my final recommendation.

---

### Official Review · Reviewer_Gqq4 · 2023-07-10

**Soundness:** 3 good
**Presentation:** 3 good
**Contribution:** 3 good
**Rating:** 5
**Confidence:** 4

**Summary:**

This paper proposes a simple yet effective approach to visualize deep learning features without using a parametric image prior. Improving upon the prior work that uses the Fourier domain formulation, the proposed method uses the polar form and optimizes the image phase to maximize the activation of specified neurons. The rest of the pipeline is similar to the prior work in terms of the optimization procedure and the transparency mask estimation. The paper further designs two new metrics for feature visualizations, where the proposed method outperforms the prior arts. In addition to feature visualization, the proposed method is applied to feature inversion of single images or learned concepts.

**Strengths:**

- The paper proposes a clever idea to optimize the phase of the image, which is commonly used for image editing, to make it look more natural for feature visualization.

- The qualitative result comparison clearly shows the significant improvement of the proposed method over the prior arts without using parametric image priors.

- The paper proposes two new metrics to compare feature visualization methods from different perspectives.

- The paper is well-written and the figures are helpful for understanding.

**Weaknesses:**

Major
----
- Lack of significant novelty. The other two contributions claimed in the paper are less convincing: the second contribution is proposed in the SmoothGrad paper as acknowledged by the paper, and the third contribution on metrics isn't thorough in its analysis.

- Lack of thorough analysis of the proposed new metrics.
    - Overall, all inverted images without a parametric image prior look so different from natural images. Thus, it's unclear if it's meaningful to show which method is slightly better via the plausible and FID metric. It's different from using FID to judge image generation or style transfer results.
    - For the plausibility/FID metric, in addition to image appearance, the image content of feature inversion results are so different from natural images. As the proposed model aims to maximize the activation of a neuron at all spatial locations, the inverted images are a collage of discriminative object parts, which is not similar to natural images. It may be better to just maximize the neuron at the center spatial location to invert the feature into one object part.
    - For the transferability score, the experiment is only done on 500 features, while ImageNet has 1,000 classes for the classification test. It's unclear how robust and reliable is the reported result. Note that there are 200 classes of dogs in ImageNet, and thus the transferability score may be different for fine-grain categories.

- Lack of comparison on scene image datasets ,e.g. PLACES. The phase of outdoor images are significantly different from indoor images. It'll be more convincing to also benchmark on scene image datasets.

- Lack of ablation study on the augmentation of the two transformations.

Minor
----
- Lack of runtime analysis. It's unclear if the proposed method is fast to run in practice

- Lack of analysis on the random seed. It'll be good to show the diversity of results using different random seeds.

**Questions:**

Are there any failure cases of the proposed method?

---

> ### Author Rebuttal · Authors · 2023-08-08
>
> We thank the reviewer for dedicating their time and for their kind words regarding our work.
>
> ## Major weaknesses
>
> - **Lack of significant novelty. The other two contributions claimed in the paper are less convincing: the second contribution is proposed in the SmoothGrad paper as acknowledged by the paper, and the third contribution on metrics isn't thorough in its analysis.**
>
> Regarding the novelty of our work, it's essential to emphasize that (1) feature visualization on recent models is not feasible without using MACO. This tool unlocks a better understanding of these models.
>
> In addition, (2) SmoothGrad does not address or talk about feature visualization and unlike SmoothGrad, we link attribution methods with feature visualization which is non-trivial, novel, and useful for the interpretability of feature visualization
> The only overlap with SmoothGrad is in that we use an average of gradients, but unlike SmoothGrad, we do not use gradients with respect to sampled noise, but rather we use the gradients obtained at each of MACO’s optimization steps to compute the transparency masks.
>
> Moreover, (3) feature visualization is a sub-field of XAI that is lacking accessible, reproducible quantitative benchmarks. Our work introduces a group of three metrics that serve as a comprehensive benchmark, thus acknowledging the challenge of designing such metrics for a field that can often rely too much on qualitative assessments, and initiating the work towards an accessible and reliable benchmark for feature visualizations.
>
> Finally, we would like to add that the anonymously released website stands as a valuable contribution on its own.
>
> - **Lack of thorough analysis of the proposed new metrics.**
>
> 	- **Overall, all inverted images without a parametric image prior look so different from natural images. Thus, it's unclear if it's meaningful to show which method is slightly better via the plausible and FID metric. It's different from using FID to judge image generation or style transfer results.**
>
> 	- **For the plausibility/FID metric, in addition to image appearance, the image content of feature inversion results are so different from natural images. As the proposed model aims to maximize the activation of a neuron at all spatial locations, the inverted images are a collage of discriminative object parts, which is not similar to natural images. It may be better to just maximize the neuron at the center spatial location to invert the feature into one object part.**
>
> 	- **For the transferability score, the experiment is only done on 500 features, while ImageNet has 1,000 classes for the classification test. It's unclear how robust and reliable is the reported result. Note that there are 200 classes of dogs in ImageNet, and thus the transferability score may be different for fine-grain categories.**
> Regarding the metrics, we have provided a detailed response in our general message (point 1). In summary, we agree with the reviewer’s  point that each metric, taken independently, may not be perfect, hence the necessity to not consider them individually but as a set.
>
> We are not sure we understand  what the reviewer means by “maximize the neuron at the center spatial location”.  If the reviewer means optimizing with respect to a central crop of the input, please note that in that case, we would have to replace the rest of the image with an arbitrary value, significantly affecting the behavior of the model and probably making the input images OOD (since a high percentage of the pixels would be equal to some non-natural arbitrary value).
>
> - **Lack of comparison on scene image datasets, e.g. PLACES. The phase of outdoor images is significantly different from indoor images. It'll be more convincing to also benchmark on scene image datasets.**
>
> Regarding ImageNet, there are 118 classes specifically for dogs. For the benchmark, the 500 classes were chosen randomly. We encourage the reviewer to explore the website to verify the results for both fine-grained and less fine-grained categories.
>
> **- Lack of ablation study on the augmentation of the two transformations.**
>
> Much in the same way than in Olah et al., MaCo needs the proposed data augmentations to regularize the optimization procedure and allow it to converge to human-interpretable solutions.
>
> ## Minor weaknesses
>
> - **Lack of runtime analysis. It's unclear if the proposed method is fast to run in practice**
> - **Lack of analysis on the random seed. It'll be good to show the diversity of results using different random seeds.**
>
> By reducing the amount of augmentations and dividing the number of parameters to optimize by half, we obtain a technique that is faster than Olah et al. Taking for example a BEiT Base with a patch size of 16, we can generate visualizations in less than 3 minutes (on a T4 GPU on Google Collab). For much smaller networks, this time is reduced to about 20 seconds per visualization.
>
> As for the visualizations using different random seeds, we take advantage of the additional rebuttal one-page pdf to showcase some visualizations for some classes using different random seeds on different models.

---

### Author Rebuttal · Authors · 2023-08-08

We extend our gratitude to the reviewers for dedicating their time to thoroughly read and assess our paper. Their critiques have been  insightful, and we believe all their concerns can be addressed. In particular,  we appreciate that all five reviewers have converged on the following two issues (which we have diligently addressed during the rebuttal period):

- 1. Providing better motivation for the introduced metrics.
- 2. Clarifying the novelty of an interactive demo to demonstrate MACO's effectiveness without cherry-picking, while acknowledging the website is not released.

We have taken the following steps to address these concerns:

**1. Regarding the metrics (Gqq4, 6q7r, UD3m):**

As pointed out by reviewer  mBBq, "there hasn't been much progress in this direction over the last few years…" and even if “ qualitative result comparison clearly shows the significant improvement ” (Gqq4) we agree with the reviewers that qualitative evaluation is not enough. This is why we propose 3 metrics in the paper:

* Plausibility. The first metric ensures that the generated image, which maximizes a logit, for example, is not Out-of-Distribution (OOD) for the model. The need of such a metric is supported by a new article [1] (out during the review period) highlighting  that common feature visualizations use a seemingly differing path from natural images (Fig. 6); hence, the necessity to ensure that the visualization is not OOD.

* FID. The FID metric ensures that the feature visualization across all classes exhibits diversity resembling the distribution of natural images, as expected given the models were trained on such images.

* Transferability. Moreover, the transferability metric ensures that the generated image is not excessively adversarial or model-specific by making sure it activates the logits of another model trained on the same dataset to some extent.

We agree with reviewers that none of these metrics are sufficient in isolation but that, **together, they form a coherent evaluation framework** –requiring generated images to robustly maximize logit scores (transferability), while encouraging features that are both diverse with respect to the distribution (FID) and   representative of the internal paths that are typical for model (non-OOD, plausibility).

We also wish to highlight that feature visualization is a sub-field of XAI that currently completely lacks accessible, reproducible quantitative benchmarks. We acknowledge the valuable work of Borowski et al. [47], but those are psychophysical benchmarks that are hard to reproduce for the broader part of the XAI community. Although these experiments are essential for measuring the impact of new methods at its ultimate goal – helping humans understand models –, they cannot be easily used during the development phase to quantify improvements over baselines, and this is why we concentrate on the construction of a standard, quantitative benchmark that is accessible to everyone and easily reproducible. In the end, we believe that initiating an accessible, reproducible quantitative benchmark, even with limitations – designing quantitative metrics for XAI is an open and active research topic because it is notoriously non-trivial – is crucial for the XAI community working on feature visualization.

**2.( qNv3,mBBq ) Regarding the website / cherry-picking.** We believe the website to be one of this paper’s major contributions and that it can help researchers better understand the intricacies of the ImageNet1k dataset. At the time of writing, we had a hard time finding a way to release it without breaking the anonymity, thus why we opted for showcasing some screenshots. However, we understand that this contribution cannot be correctly judged without actually browsing through the website, and this is why during the review period, we have decided to develop an anonymized version of the website to showcase and demonstrate the consistency of our results across the concepts of all 1000 classes in ImageNet. The website features 10,000 visualizations of ImageNet concepts, allowing users to **view the most activated examples with a simple "click" on any feature visualization**. https://anon1285.github.io/neurips1285/

https://anon1285.github.io/neurips1285/classes/espresso/

https://anon1285.github.io/neurips1285/classes/pineapple/

https://anon1285.github.io/neurips1285/classes/fig/

https://anon1285.github.io/neurips1285/classes/orange/

https://anon1285.github.io/neurips1285/classes/bubble/

https://anon1285.github.io/neurips1285/classes/admiral/

https://anon1285.github.io/neurips1285/classes/ostrich/

https://anon1285.github.io/neurips1285/classes/jellyfish/

https://anon1285.github.io/neurips1285/classes/volcano/

https://anon1285.github.io/neurips1285/classes/honeycomb/

In addition, we provide some additional visualizations on the one-page rebuttal PDF for two different models (BEiT and ResNet50V2) for 4 different classes using 4 different random seeds.

Moreover, we have incorporated the necessary citations and clarifications into our Related Work and Discussion sections. We will address each reviewer’s remaining comments directly.

Finally, we wish to underscore that our work has successfully unlocked feature visualization for modern large scale deep networks through a straightforward and well-motivated adjustment of the optimization problem. This achievement holds potential implications for all studies seeking to understand deep networks using feature visualization. We express our sincere gratitude to the reviewers for their valuable suggestions and remarks, which have further solidified the evidence supporting our findings.

[1] “Don’t trust your eyes: on the (un)reliability of feature visualizations.” Geirhos et al., arXiV.

---

### Decision · Program_Chairs · 2023-09-21

**Decision:**

Accept (poster)

**Comment:**

The reviewers did not reach a consensus and are clustered around the borderline. We believe this paper brings more useful information to the community than not and hence accept this paper. For instance, making visualizations work for modern networks and the website itself are two worthy contributions. The lack of user study is indeed a downside, but we believe the pros outweigh the cons here.